# Reinforcement Learning with Imperfect Transition Predictions: A Bellman-Jensen Approach

**Chenbei Lu**
Institute for Interdisciplinary Information Sciences
Tsinghua University

**Zaiwei Chen**
Edwardson School of Industrial Engineering
Purdue University

**Tongxin Li**
School of Data Science
The Chinese University of Hong Kong (Shenzhen)

**Chenye Wu**[*]
School of Science and Engineering
The Chinese University of Hong Kong (Shenzhen)

**Adam Wierman**
Computing & Mathematical Sciences
Caltech

## Abstract

Traditional reinforcement learning (RL) assumes the agents make decisions based on Markov decision processes (MDPs) with one-step transition models. In many real-world applications, such as energy management and stock investment, agents can access multi-step predictions of future states, which provide additional advantages for decision making. However, multi-step predictions are inherently high-dimensional: naively embedding these predictions into an MDP leads to an exponential blow-up in state space and the curse of dimensionality. Moreover, existing RL theory provides few tools to analyze prediction-augmented MDPs, as it typically works on one-step transition kernels and cannot accommodate multi-step predictions with errors or partial action-coverage. We address these challenges with three key innovations: First, we propose the *Bayesian value function* to characterize the optimal prediction-aware policy tractably. Second, we develop a novel *BellmanJensen Gap* analysis on the Bayesian value function, which enables characterizing the value of imperfect predictions. Third, we introduce BOLA (Bayesian Offline Learning with Online Adaptation), a two-stage model-based RL algorithm that separates offline Bayesian value learning from lightweight online adaptation to real-time predictions. We prove that BOLA remains sample-efficient even under imperfect predictions. We validate our theory and algorithm on synthetic MDPs and a real-world wind energy storage control problem.

## 1 Introduction

Reinforcement Learning (RL) [1] has emerged as a powerful framework for sequential decision-making, achieving remarkable success across diverse domains [2–5]. Classical RL formulates

---

[*]Corresponding author, `chenyewu@yeah.net`

39th Conference on Neural Information Processing Systems (NeurIPS 2025).

decision-making as a Markov Decision Process (MDP), where an agent seeks to maximize expected cumulative rewards in a stochastic environment. A central premise of this framework is that once the agent has accurately captured the environment model (e.g., the transition dynamics), it can, in principle, compute an optimal policy. In model-based RL, this involves explicitly learning the transition kernel and reward function to solve the optimal policy [6]. Even in model-free RL, agents implicitly learn the environment through value function or policy learning [7].

However, in many real-world applications, agents can access even richer information: the prediction of future transition realizations. Rather than relying solely on expected transition dynamics, these realization-level predictions specify exact future states, which can reduce or even eliminate the environments inherent stochasticity and enable more effective decision-making. For example, in financial markets, accurate multi-step price forecasts can substantially improve trading strategies [8], while in energy systems, reliable predictions of renewable energy generation allow the system operators to schedule the power generation sources more efficiently [9].

Despite their potential, incorporating multi-step transition predictions into MDPs faces three key challenges. First, the predictions over a multi-step horizon are inherently high-dimensional: augmenting the state with these predictions expands the state space exponentially, making standard solutions computationally intractable (see Section 3 for more details). Second, even if the augmented MDP can be solved, existing theory lacks formal tools to quantify the benefits of multi-step transition predictions, particularly when they are inaccurate or only cover a subset of actions. Third, in the absence of strong assumptions on function approximation [1, 10–13], RL's sample complexity scales at least linearly with the size of the stateaction space [14, 15], and the exponential state expansion also induces an exponential blow-up in the required samples [16]. Addressing these challenges is essential for the rigorous integration of transition predictions into RL. See Appendix A for a detailed literature review.

To overcome these challenges, our contributions can be summarized as follows:

**Tractable Optimal Policy for MDPs with Transition Predictions.** We introduce a low-dimensional Bayesian value function that integrates multi-step transition predictions into the value evaluation, which enables a tractable characterization of the optimal prediction-aware policy.

**Characterization of the Value of Imperfect Predictions using Bellman-Jensen Gap.** We introduce the Bellman-Jensen Gap framework, a novel analytical tool that decomposes the advantage of multi-step predictions into a recursive sum of local Jensen gaps in the Bayesian value function. Building on this framework, we characterize the value of imperfect predictions and show how it can close the performance gap to the offline optimal policy in Theorem 4.1.

**Prediction-Aware Algorithm with Improved Sample Complexity.** We propose BOLA, a two-stage model-based RL algorithm that combines offline Bayesian value estimation with online integration of real-time predictions. We prove that BOLA avoids the exponential sample complexity and, given high-quality predictions, is more sample efficient than classical model-based RL [14, 15]. Our analysis relies on tailored error-decomposition and telescoping bounds to control multi-step transition errors.

The remainder of this paper proceeds as follows. Section 2 formalizes the prediction-augmented MDP framework. Based on this formulation, Section 3 introduces a tractable Bayesian value function to characterize the optimal policy, circumventing the curse of dimensionality. Subsequently, Section 4 establishes the Bellman-Jensen Gap to theoretically quantify the value of imperfect predictions. This analysis directly motivates Section 5, where we present the BOLA algorithm with provable sample efficiency guarantees. Section 6 then provides empirical validation on a wind energy storage problem. Finally, Section 7 concludes with a discussion of limitations and future work. Complete proofs and additional details are included in the appendices.

## 2 Markov Decision Processes with Transition Predictions

In this section, we formally introduce the framework for MDPs augmented with transition predictions.

We begin by introducing the definition of a discounted infinite-horizon MDP, which is specified by the tuple $\mathcal{M} = (\mathcal{S}, \mathcal{A}, P, r, \gamma)$. Here, $\mathcal{S}$ and $\mathcal{A}$ are the finite state and action spaces, respectively; $P$ is the transition kernel, i.e., $P(\cdot \mid s, a)$ denotes the distribution of the next state given that action $a$

is taken at state $s$; $r : \mathcal{S} \times \mathcal{A} \rightarrow [0, 1]$ is the reward function; and $\gamma \in (0, 1)$ is the discount factor. Since we focus on finite MDPs, assuming bounded rewards is without loss of generality. The model parameters of the MDP (i.e., the transition kernel $P$ and the reward function $r$) are unknown to the agent, but the agent can interact with the environment by observing the current state, selecting actions, and receiving the resulting next state and reward.

## 2.1 MDPs with Transition Predictions

We extend the classical MDP framework by incorporating imperfect predictions of future transition dynamics. Formally, consider an MDP with transition predictions characterized by the tuple $\mathcal{M}_p = (\mathcal{S}, \mathcal{A}, P, r, \gamma, K, \mathcal{A}^-, \varepsilon)$, where $(K, \mathcal{A}^-, \varepsilon)$ capture the prediction structure. Specifically, $K$ denotes the finite prediction horizon; $\mathcal{A}^- \subseteq \mathcal{A}$ specifies the subset of actions for which transition outcomes can be predicted; and $\varepsilon$ quantifies the associated prediction errors.

We first consider an ideal case that the prediction is accurate. At discrete time steps $t = 0, K, 2K, \ldots$, the agent receives a batch of predicted transitions for the next $K$ steps, denoted as $\boldsymbol{\sigma}^* = (\sigma_1^*, \sigma_2^*, \ldots, \sigma_K^*)$, and each $\sigma_k^*$ is a binary matrix of size $|\mathcal{S}||\mathcal{A}^-| \times |\mathcal{S}|$, where each row corresponds to a $(s, a)$ pair with $a \in \mathcal{A}^-$ and is a one-hot vector indicating the predicted next state.

Given an accurate one-step transition prediction $\sigma_k^*$, the conditional transition probabilities depend on whether the action taken falls within the predictable action subset $\mathcal{A}^-$. This subset captures actions for which reliable prediction models are available, allowing the agent to exploit future information. In contrast, actions outside $\mathcal{A}^-$ must rely solely on the underlying transition dynamics $P(s'|s, a)$ of the environment. Accordingly, the conditional transition model with predictions is:

$$P(s' \mid s, a, \sigma_k) = \begin{cases} \sigma_k^*((s, a), s'), & \forall a \in \mathcal{A}^-, s, s' \in \mathcal{S}, \\ P(s' \mid s, a), & \forall a \notin \mathcal{A}^-, s, s' \in \mathcal{S}. \end{cases} \tag{1}$$

To preserve the Markov property of the underlying MDP, we impose two natural conditions on each onestep prediction matrix $\sigma_k^*$ within the $K$-step forecast:

- *Independence and stationarity*. Each $\sigma_k^*$ is drawn i.i.d. from a fixed distribution. This ensures that every predicted transition remains stationary with respect to the transition kernel. Also, the prediction depends only on the current stateaction pair and not on any prior history.

- *Consistency*. In expectation, the accurate prediction exactly recovers the true transition kernel:

$$\mathbb{E}_{\sigma_k^* \sim P_{\sigma^*}} \left[ \sigma_k^*((s, a), s') \right] = P(s' \mid s, a), \quad \forall k \leq K, a \in \mathcal{A}^-, s, s' \in \mathcal{S}. \tag{2}$$

However, exact prediction is not always attainable in practice. Thus, we assume the agent only receives inaccurate predictions denoted as $\boldsymbol{\sigma} = (\sigma_1, \sigma_2, \ldots, \sigma_K) \in \mathcal{Q}_K$, where $\mathcal{Q}_K$ denotes the space of inaccurate prediction $\boldsymbol{\sigma}$, and each $\sigma_k \in [0, 1]^{|\mathcal{S}||\mathcal{A}^-| \times |\mathcal{S}|}$ is a stochastic matrix indicating predicted transition probabilities for the state-action pairs at step $k$ in the future. Each prediction $\sigma_k$ may differ from the true future transition $\sigma_k^*$ due to prediction error. We model this discrepancy as $\sigma_k = \sigma_k^* + \varepsilon_k, k = 1, \ldots, K$, where $\varepsilon_k$ is a random error matrix drawn from a distribution $f_{\varepsilon_k | \sigma_k^*}$.

This formulation captures a broad class of imperfect predictions, allowing us to study how finite-horizon, partial, and inaccurate forecasts can be leveraged for improved decision-making in MDPs.

**Remark:** Our model differs fundamentally from prior work such as [17, 18], which treats predictions as noisy estimates of the transition kernel and aims only to match standard MDP performance. In contrast, we model ideal predictions as concrete, one-hot realizations with certain consistency property in Eq. (2), enabling us to leverage realization-level information to *surpass* classic MDP performance. Furthermore, unlike these methods, we explicitly address *partial action predictability*, an open challenge identified in [17].

## 2.2 Decision-Making and Optimization Objectives

We adopt a fixed-horizon planning protocol [19], where the agent makes decisions at discrete time points $t = 0, K, 2K, \ldots$. At each decision point, after observing the current state $s_t$ and receiving the prediction batch $\boldsymbol{\sigma}$, the agent selects an action sequence $\boldsymbol{a} = (a_0, \ldots, a_{K-1}) \in \mathcal{A}^K$ according to a

policy $\pi : \mathcal{S} \times \mathcal{Q}_K \to \Delta(\mathcal{A}^K)$, where $\mathcal{Q}_K$ denotes the prediction space and $\Delta(\mathcal{A}^K)$ denotes the probability simplex over $K$-step action sequences. This setting models how agents dynamically plan decisions over a finite prediction horizon. The agent seeks to maximize the expected cumulative reward by selecting an optimal policy, defined as $\pi^* = \arg\max_\pi \mathbb{E}_\pi \left[ \sum_{t=0}^\infty \gamma^t r(s_t, a_t) \,|\, s_0 = s, \boldsymbol{\sigma}_0 = \boldsymbol{\sigma} \right]$ for all $s \in \mathcal{S}, \boldsymbol{\sigma} \in \mathcal{Q}_K$.

# 3 Bayesian Value Function and Prediction-Aware Optimal Policy

In this section, we develop a tractable formulation for decision-making with multi-step imperfect transition predictions.

**Motivation:** A straightforward strategy for using predictions is to treat $(s, \boldsymbol{\sigma})$ as the new state and solve a standard MDP over this extended state space. However, this approach quickly becomes intractable. A $K$-step prediction $\boldsymbol{\sigma} = (\sigma_1, \ldots, \sigma_K)$ consists of $K$ transition matrices, each of size $|\mathcal{S}||\mathcal{A}| \times |\mathcal{S}|$, resulting in an exponentially large state space size of at least $|\mathcal{S}|^{K|\mathcal{S}||\mathcal{A}|}$. Moreover, since $\boldsymbol{\sigma}$ is typically noisy and continuous, its support can be uncountably infinite. As a result, the augmented value function must satisfy an infinite-dimensional Bellman equation, making classical solutions impractical even for $K = 1$.

In summary, while predictions have the potential to improve performance, naively augmenting the state space with raw prediction vectors leads to intractable computation. To address this issue, we next introduce a Bayesian value function that enables a tractable, prediction-aware characterization of the optimal policy.

## 3.1 Bayesian Value Function and Optimal Policy Structure

To avoid explicit state augmentation, we instead formulate a *Bayesian value function* defined over the original state space. The key idea is to take an expectation over the prediction distribution, thereby shifting the complexity into an outer integral while preserving a tractable structure. Formally, we define the Bayesian value function as:

$$V_{K,\mathcal{A}^-,\varepsilon}^{\text{Bayes},\pi}(s) := \mathbb{E}_{\boldsymbol{\sigma}} \left[ \mathbb{E}_\pi \left[ \sum_{t=0}^\infty \gamma^t r(s_t, a_t) \,\Big|\, s_0 = s, \boldsymbol{\sigma}_0 = \boldsymbol{\sigma} \right] \right]. \tag{3}$$

This Bayesian value function represents the expected cumulative reward when each decision is made after drawing a $K$-step prediction $\boldsymbol{\sigma}$. We call it Bayesian because we marginalize over the distribution of $\boldsymbol{\sigma}$, thereby accounting for forecast uncertainty in the value estimate. Importantly, the policy can condition on the realized $\boldsymbol{\sigma}$, yet the value function itself remains defined solely over the original state space. This preserves tractability by avoiding an explicit statespace augmentation. The optimal Bayesian value is then:

$$V_{K,\mathcal{A}^-,\varepsilon}^{\text{Bayes},*}(s) = \max_\pi V_{K,\mathcal{A}^-,\varepsilon}^{\text{Bayes},\pi}(s), \quad \forall s \in \mathcal{S}. \tag{4}$$

By constructing an auxiliary MDP that incorporates the predictions and linking it with the optimal Bayesian value function, we derive the corresponding Bellman optimality equation (see Appendix B for the proof).

**Theorem 3.1** (Bellman Optimality Equation for Bayesian Value Function). *The optimal Bayesian value function $V_{K,\mathcal{A}^-,\varepsilon}^{\text{Bayes},*}$ is the unique solution to the following fixed-point equation:*

$$V_{K,\mathcal{A}^-,\varepsilon}^{\text{Bayes},*}(s) = \mathbb{E}_{\boldsymbol{\sigma}} \left[ \max_{\boldsymbol{a}} \left( \sum_{t=0}^{K-1} \gamma^t \left( \sum_{s_t} P(s_t|s, \boldsymbol{a}_{0:t-1}, \boldsymbol{\sigma}_{1:t}) r(s_t, a_t) \right) \right. \right.$$
$$\left. \left. + \gamma^K \sum_{s_K} P(s_K|s, \boldsymbol{a}, \boldsymbol{\sigma}) V_{K,\mathcal{A}^-,\varepsilon}^{\text{Bayes},*}(s_K) \right) \right], \ \forall s \in \mathcal{S}. \tag{5}$$

*Here in Eq. (5), $\boldsymbol{a}_{0:t-1} = (a_0, \ldots, a_{t-1})$ and $\boldsymbol{\sigma}_{1:t} = (\sigma_1, \ldots, \sigma_t)$ denote the sequences of actions and predictions, respectively, and $P(s_t|s_0, \boldsymbol{a}_{0:t-1}, \boldsymbol{\sigma}_{1:t})$ is the multi-step transition probability from initial state $s_0$ to state $s_t$ after $t$ steps under the sequences of actions and predictions $\boldsymbol{a}_{0:t-1}$ and $\boldsymbol{\sigma}_{1:t}$, which satisfies the following recursive relation:*

$$P(s_t|s_0, \boldsymbol{a}_{0:t-1}, \boldsymbol{\sigma}_{1:t}) = \sum_{s_{t-1} \in \mathcal{S}} P(s_t|s_{t-1}, a_{t-1}, \sigma_t) P(s_{t-1}|s_0, \boldsymbol{a}_{0:t-2}, \boldsymbol{\sigma}_{1:t-1}), \forall t. \tag{6}$$

The recursive form in Eq. (5) captures how predictions guide near-term planning over horizon $K$, with long-term value rolled into $V_{K,\mathcal{A}^-,\varepsilon}^{\text{Bayes},*}(s_K)$. Importantly, the corresponding Bellman operator is a contraction mapping with parameter $\gamma^K$ under the infinity norm, which guarantees the existence and uniqueness of the solution and enables efficient fixed-point computation.

The optimal Bayesian value function directly yields the optimal policy, as characterized below. The proof is provided in Appendix C.

**Corollary 3.1** (Optimal Policy with Bayesian Value Function and Transition Predictions). *The optimal policy $\pi^*(\cdot \mid s, \boldsymbol{\sigma})$ with $K$-step transition predictions $\boldsymbol{\sigma}$ satisfies:*

$$\{\boldsymbol{a} \in \mathcal{A}^K \mid \pi^*(\boldsymbol{a} \mid s, \boldsymbol{\sigma}) > 0\}$$

$$\subseteq \arg\max_{\boldsymbol{a} \in \mathcal{A}^K} \left( \sum_{t=0}^{K-1} \gamma^t \sum_{s_t} P(s_t \mid s, \boldsymbol{a}_{0:t-1}, \boldsymbol{\sigma}_{1:t}) r(s_t, a_t) + \gamma^K \sum_{s_K} P(s_K \mid s, \boldsymbol{a}, \boldsymbol{\sigma}) V_{K,\mathcal{A}^-,\varepsilon}^{\text{Bayes},*}(s_K) \right)$$

$$\forall s \in \mathcal{S}, \boldsymbol{\sigma} \in \mathcal{Q}_K. \tag{7}$$

This result shows that the optimal prediction-aware policy can be computed via a finite-horizon planning over $\boldsymbol{\sigma}$, followed by terminal reward using the Bayesian value function $V_{K,\mathcal{A}^-,\varepsilon}^{\text{Bayes},*}$. In effect, we have reduced the original infinite-horizon problem with high-dimensional predictions to a special form of fixedhorizon planning [19], which is tractable without explicitly augmenting the state space.

# 4 Analyzing the Value of Predictions

In this section, we examine how access to transition predictions improves decision-making in MDPs. Classical MDPs face a structural limitation: their value functions involve deeply nested $\max$-over-$\mathbb{E}$ operations, which force agents to commit to fixed policies based on expected dynamics. We show that transition predictions alleviate this limitation by enabling a localized reordering of the $\max$ and $\mathbb{E}$ operators, allowing actions to adapt to realized transitions. We use a *Bellman-Jensen Gap* analysis on the Bayesian value function to characterize the value of predictions.

## 4.1 Bellman-Jensen Gap

We introduce the Bellman-Jensen Gap by comparing the following value functions.

**Bellman Expansion of Optimal Value Function.** By recursively applying the Bellman optimality equation for classical discounted MDPs, the value function can be expressed in the following nested form [20]:

$$V_{\text{MDP}}^*(s_0) = \max_{a_0} \left[ r(s_0, a_0) + \gamma \mathbb{E}_{\sigma_1^*} \left[ \max_{a_1} \left[ r(s_1, a_1) + \gamma \mathbb{E}_{\sigma_2^*} \left[ \max_{a_2} [r(s_2, a_2) + \cdots] \right] \right] \right] \right], \tag{8}$$

where $\sigma_t^*$ denotes the transition realization of $s_t$ at time $t$. Each expectation $\mathbb{E}_{\sigma_t^*}$ is equivalent to taking the expectation over the next state $s_t \sim P(\cdot \mid s_{t-1}, a_{t-1})$, corresponding to the transition dynamics governed by $\sigma_t^*$. This formulation results in a deeply nested $\max$-over-$\mathbb{E}$ structure, where the agent must choose an action that is optimal in expectation, without the ability to anticipate and adapt to future information.

In contrast, if the agent had access to perfect predictions of future transitions, it could defer action selection until those transitions are known, which allows a *localized reordering* of the $\max$ and $\mathbb{E}$ operators. We use a one-step prediction case to illustrate it:

**Operator Reordering with One-Step Prediction.** Recall the Bayesian value function with $K = 1$, which can be expanded into the following recursive form:

$$V_{K=1,\mathcal{A},\mathbf{0}}^{\text{Bayes},*}(s_0) = \mathbb{E}_{\sigma_1^*} \left[ \max_{a_0} \left[ r(s_0, a_0) + \gamma \mathbb{E}_{\sigma_2^*} \left[ \max_{a_1} [r(s_1, a_1) + \cdots] \right] \right] \right]. \tag{9}$$

Observe that, with one-step prediction, each $\mathbb{E}_{\sigma_t^*}$ operator is moved to the outer side of the neighborhood $\max_{a_{t-1}}$ operator. Intuitively, it provides the agent the ability to make decisions according to the transition prediction $\sigma_t^*$ at time $t$. Mathematically, this localized reordering creates a *local*

*Jensen gap* by exploiting Jensen's inequality due to $\mathbb{E}_\sigma[\max_a f(s, a; \sigma)] \geq \max_a \mathbb{E}_\sigma[f(s, a; \sigma)]$, where discrete maximization is a convex function, and $f(s, a; \sigma)$ denotes the expected return under state $s$ with action $a$ and prediction $\sigma$. Since the Bayesian value function contains infinitely many such operator reordering in a recursive manner, we term it as the *Bellman-Jensen Gap*.

**Maximal Bellman-Jensen Gap with Infinite-Step Prediction.** Such Bellman-Jensen Gaps reach the maximum when the prediction horizon is infinite. Formally, let $V_{\mathrm{off}}^{\mathrm{Bayes},*} \in \mathbb{R}^{|\mathcal{S}|}$ denote the offline optimal Bayesian value function, where the agent has exact knowledge of all future transitions:

$$V_{\mathrm{off}}^{\mathrm{Bayes},*}(s_0) = \mathbb{E}_{\boldsymbol{\sigma}_1^*} \mathbb{E}_{\boldsymbol{\sigma}_2^*} \cdots \left[ \max_{a_0} \left[ r(s_0, a_0) + \gamma \max_{a_1} \left[ r(s_1, a_1) + \gamma \max_{a_2} [r(s_2, a_2) + \cdots] \right] \right] \right]$$

$$= \lim_{k \to \infty} \mathbb{E}_{\boldsymbol{\sigma}_{1:k}^*} \left[ \max_{\boldsymbol{a}_{0:k-1}} \left[ \sum_{t=0}^{k-1} \gamma^t r(s_t, a_t) \right] \right], \tag{10}$$

where $\boldsymbol{\sigma}_{1:k}^*$ is the sequence of realized transition kernels and $\boldsymbol{a}_{0:k-1}$ is the action sequence over horizon $k$. The existence of the limit is shown in Appendix D.

Observe that, with infinitely long accurate prediction, all $\mathbb{E}_{\sigma^*}$ operators appear outside of any $\max_a$ operator, which indicates that the agent can make the decision with full information of all future information, yielding the maximal Bellman-Jensen Gap defined as follows:

**Definition 4.1** (Maximal Bellman-Jensen Gap). *For any state $s \in \mathcal{S}$, we define the* Maximal Bellman-Jensen Gap *as $\Delta(s) := V_{\mathrm{off}}^{\mathrm{Bayes},*}(s) - V_{\mathrm{MDP}}^*(s)$, which quantifies the greatest possible performance gain from knowing exact future transitions.*

The maximal Bellman-Jensen Gap characterizes the fundamental benefit that predictive information can offer in MDPs. It upper-bounds the value of any prediction by capturing the intrinsic benefits of operator reordering in the value function. Note that, this analytical framework naturally extends to other types of predictions. For example, by redefining $\sigma$ to represent the prediction on reward realizations, the same Bellman-Jensen Gap analysis applies.

## 4.2 Closing the BellmanJensen Gap with Imperfect Predictions

We now leverage the BellmanJensen gap framework to analyze how imperfect predictions narrow the performance gap to the offline oracle. In particular, we derive explicit bounds on the suboptimality of a policy that uses finite-horizon, inaccurate, and partial action-coverage predictions.

The following theorem provides a finite-horizon performance bound that decomposes the suboptimality into three interpretable components, each capturing a distinct structural limitation. The proof is provided in Appendix E.

**Theorem 4.1** (Bellman-Jensen Performance Bound). *Given any prediction with horizon $K \geq 1$, predictable action set $\mathcal{A}^- \subseteq \mathcal{A}$ and prediction errors $\boldsymbol{\varepsilon}$, the performance gap between the prediction-aware policy and the offline optimal policy satisfies:*

$$\max_{s \in \mathcal{S}} \left( V_{\mathrm{off}}^{\mathrm{Bayes},*}(s) - V_{K, \mathcal{A}^-, \boldsymbol{\varepsilon}}^{\mathrm{Bayes},*}(s) \right) \leq \underbrace{\frac{C_1 \gamma^K \sqrt{K \log |\mathcal{A}|}}{(1-\gamma)^{\frac{6}{5}}(1-\gamma^{2K})}}_{A_1 : \text{loss due to finite prediction window}} + \underbrace{\sum_{j=1}^{K} \frac{\gamma^j}{(1-\gamma)(1-\gamma^K)} \epsilon_j}_{A_2 : \text{loss due to prediction error}}$$

$$+ \underbrace{C_2 \sum_{t=1}^{\infty} \gamma^t \sqrt{\log(|\mathcal{A}|^{t+1} - |\mathcal{A}^-|^{t+1} + 1)\theta_{\max}^2}}_{A_3 : \text{loss due to partial action predictability}},$$

*where $C_1$ and $C_2$ are absolute constants, $\epsilon_j$ denotes the prediction error at step $j$, defined as the Wasserstein-1 distance between the predicted and true transition distributions; parameter $\theta_{\max}^2 = \max_{s, \boldsymbol{a}_{0:t}, t} \sigma^2(r(s_t, a_t | s_0 = s, \boldsymbol{a}_{0:t}))$ captures the variability of the reward, and $\sigma(\cdot)$ denotes the sub-Gaussian parameter.*

**Interpretation.** Theorem 4.1 shows that the performance gap decomposes into three terms. The first term $A_1$ quantifies the performance loss due to the finite prediction horizon $K$. The factor $\gamma^K$ reflects that the benefit of predictions decreases exponentially with the horizon length, implying that even short-term predictions can capture significant potential improvement. When $K \to \infty$, this loss term

diminishes. The $(1 - \gamma)^{6/5}$ exponent arises from a refined dyadic horizon decomposition argument controlling the dependence on the discount factor (see Lemmas E.2 and E.3 for details). The term $\sqrt{\log|\mathcal{A}|}$ shows the number of actions slightly increases this gap, as a larger action space makes it statistically harder to identify the optimal action under uncertainty.

The second term $A_2$ captures the impact of prediction errors and disappears as the predicted transitions become accurate. Notably, it highlights that errors in subsequent steps have progressively smaller effects on overall performance, aligning with practical intuition. When $\epsilon_j = \epsilon$ for all $j$, we have $A_2 = \mathcal{O}(\epsilon/(1-\gamma)^2)$, which is independent of $K$, indicating that this term is primarily governed by the average prediction error, rather than the length of the prediction horizon.

The third term $A_3$ arises from partial action predictability and vanishes when all actions are predictable (i.e., $\mathcal{A}^- = \mathcal{A}$). It is scaled by $\sqrt{\theta_{\max}^2}$, indicating that greater reward uncertainty amplifies the Bellman-Jensen Gap. When $\mathcal{A}^- = \emptyset$, this term will not blow up and simplifies to $\mathcal{O}(\sqrt{\log|\mathcal{A}|\theta_{\max}^2}(1-\gamma)^{-\frac{3}{2}})$.

**Corollary 4.1.** *Given any prediction horizon $K \geq 1$, if the predictions are perfectly accurate with $\epsilon_j = 0$ for all $1 \leq j \leq K$, and all actions are predictable with $\mathcal{A}^- = \mathcal{A}$, then the maximal performance gap satisfies $\max_{s \in \mathcal{S}}(V_{\text{off}}^{\text{Bayes},*}(s) - V_{K,\mathcal{A}^-,\varepsilon}^{\text{Bayes},*}(s)) \leq \mathcal{O}(\gamma^K \sqrt{K})$.*

This result demonstrates that sufficiently accurate predictive informationeven over a finite horizoncan dramatically reduce the fundamental Bellman-Jensen Gap, bringing the agents performance significantly closer to the offline oracle benchmark. It characterizes the theoretical upper bound on the improvement that predictive signals can offer, revealing an exponential decay in the gap with horizon length $K$, up to a sublinear $\sqrt{K}$ correction term.

# 5 BOLA: Bayesian Offline Learning with Online Adaptation

Building on the theoretical understanding of the prediction-aware policy, in this section, we present a practical model-based algorithm for implementing the prediction-aware optimal policy.

The key insight from Theorem 3.1 and Corollary 3.1 is that optimal decisions can be achieved by combining short-horizon planning with a precomputed Bayesian value as the terminal function, which can effectively leverage predictive information without explicitly expanding the state space. This motivates the design of **BOLA**, a two-stage approach that cleanly separates offline learning from online adaptation to predictions.

## 5.1 BOLA Algorithm Overview

We propose **BOLA** (Bayesian Offline Learning with Online Adaptation), a model-based reinforcement learning algorithm designed to exploit transition predictions for efficient decision-making. BOLA decomposes learning and planning into two stages: (1) **Offline Stage:** Estimate the Bayesian value function $V_{K,\mathcal{A}^-,\varepsilon}^{\text{Bayes},*}(s)$ from samples by solving the Bellman equation in Eq. (5), and (2) **Online Stage:** At each decision point, observe real-time transition predictions $\boldsymbol{\sigma}$ and compute the optimal short-horizon action sequence using Eq. (7).

**Offline Bayesian Value Function Learning.** To implement the prediction-aware Bellman operator from Eq. (5), we adopt a model-based learning approach inspired by classical MDPs. Specifically, we estimate the key quantities required to compute the Bayesian value function $V_{K,\mathcal{A}^-,\varepsilon}^{\text{Bayes},*}(s)$ via value iteration. These include: (1) the reward function $r(s,a)$, (2) the distribution over $K$-step transition predictions $P(\boldsymbol{\sigma})$, and (3) the multi-step transition kernel $P(s' \mid s, \boldsymbol{a}, \boldsymbol{\sigma})$.

Importantly, the recursive structure in Eq. (6) allows us to avoid estimating the full $K$-step transition model directly. Instead, it suffices to estimate one-step transition probabilities $P(s' \mid s, a, \sigma)$ for predictable actions $a \in \mathcal{A}^-$, and standard MDP transitions $P(s' \mid s, a)$ for actions $a \notin \mathcal{A}^-$.

We assume access to a generative model [21, 22]. For each state-action pair $(s, a)$ with $a \notin \mathcal{A}^-$, the generative model allows us to generate $N_1$ independent next-state samples, denoted by $\{s_{(s,a)}^i\}_{i=1}^{N_1}$.

For estimating the prediction distribution, we assume there exists a *prediction oracle* defined as follows:

**Assumption 5.1** (Prediction Oracle). *The agent has access to a prediction oracle $\mathfrak{D}_{\mathrm{pred}}$ that, upon query, returns independent samples $\boldsymbol{\sigma}^i \sim P(\boldsymbol{\sigma})$, where $\boldsymbol{\sigma}^i = (\sigma_1^i, \ldots, \sigma_K^i)$ represents a $K$-step transition prediction vector drawn from the underlying distribution $P(\boldsymbol{\sigma})$.*

Under this assumption, we draw $N_2$ independent prediction samples from the oracle, denoted by $\{\boldsymbol{\sigma}^i = (\sigma_1^i, \ldots, \sigma_K^i)\}_{i=1}^{N_2}$, which are then used to estimate $P(\boldsymbol{\sigma})$ via empirical frequencies.

Using these samples, we estimate the relevant probabilities by empirical frequency:

$$\widehat{P}(s' \mid s, a) = \frac{1}{N_1} \sum_{i=1}^{N_1} \mathbb{1}(s_{(s,a)}^i = s'), \quad \forall a \in \mathcal{A} \setminus \mathcal{A}^-, \tag{11}$$

$$\widehat{P}(\boldsymbol{\sigma}) = \frac{1}{N_2} \sum_{i=1}^{N_2} \mathbb{1}(\boldsymbol{\sigma}^i = \boldsymbol{\sigma}), \quad \forall \boldsymbol{\sigma} \in \mathcal{Q}_K. \tag{12}$$

The reward function $r(s, a)$ is obtained by sampling from each state-action pair $(s, a)$ once.

With all model components estimated, we apply value iteration on the prediction-augmented Bellman operator in Eq. (5) to compute an approximate Bayesian value function $\widehat{V}_{K,\mathcal{A}^-,\varepsilon}^{\mathrm{Bayes},*}$. The contraction property of the Bellman operator ensures that this fixed-point iteration converges.

**Online Adaptation.** At each decision point, BOLA receives a $K$-step transition prediction vector $\boldsymbol{\sigma} = (\sigma_1, \ldots, \sigma_K)$. Given the current state $s$, BOLA first evaluates the multi-step transition probabilities by incorporating the prediction $\boldsymbol{\sigma}$, and then solves the optimal action sequence through Eq. (7) using the precomputed Bayesian value function $\widehat{V}_{K,\mathcal{A}^-,\varepsilon}^{\mathrm{Bayes},*}$.

This scheme enables real-time adaptation without solving high-dimensional value functions online. By combining offline long-term terminal value estimation with short-horizon prediction-aware planning [23], BOLA avoids state space augmentation and maintains computational tractability. Algorithm 1 summarizes the BOLA procedure.

---

**Algorithm 1** BOLA: Bayesian Offline Learning with Online Adaptation

---

1: Sample $N_1$ times from each $(s, a) \in \mathcal{S} \times \mathcal{A} \setminus \mathcal{A}^-$ and get samples $\{s_{(s,a)}^i\}_{i=1}^{N_1}$;
2: Sample $N_2$ empirical predictions $\{\boldsymbol{\sigma}^i\}_{i=1}^{N_2}$ from the prediction oracle $\mathfrak{D}_{\mathrm{pred}}$;
3: Estimate transition model $\widehat{P}(s' \mid s, a)$ and distribution $\widehat{P}(\boldsymbol{\sigma})$ according to Eq. (11)-(12);
4: Solve the estimated Bayesian value function $\widehat{V}_{K,\mathcal{A}^-,\varepsilon}^{\mathrm{Bayes},*}$ using value iteration;
5: **for** each decision timestep **do**
6:     Observe the current state $s$ and the transition prediction $\boldsymbol{\sigma}$;
7:     **for** $t = 1$ to $K$ **do**
8:         Compute the transition probabilities $P(s_t \mid s, \boldsymbol{a}_{0:t-1}, \boldsymbol{\sigma}_{1:t})$ using Eq. (6);
9:     **end for**
10:     Determine the optimal policy $\pi^*(\cdot \mid s, \boldsymbol{\sigma})$ by solving Eq. (7);
11: **end for**

---

## 5.2 Sample Complexity Guarantees

This section presents the sample complexity guarantees of Algorithm 1, more specifically, the learning of the Bayesian value function. The proof of the following theorem is presented in Appendix F.

**Theorem 5.1.** *For any given MDP, any confidence level $\delta \in (0, 1)$, any desired accuracy level $\epsilon \in (0, \frac{1}{1-\gamma})$, a tradeoff parameter $\alpha \in (0, 1)$, and prediction horizon $K \geq 1$, let $D_1$ be the number of samples drawn from the generative model, and $D_2$ be the number of samples drawn from the prediction oracle $\mathfrak{D}_{\mathrm{pred}}$. If*

$$D_1 = \frac{\overline{C}_1 |\mathcal{S}|(|\mathcal{A}| - |\mathcal{A}^-|) \log\left(K|\mathcal{S}|(|\mathcal{A}| - |\mathcal{A}^-|)/\delta\right)}{(1-\gamma)^4 (1-\alpha)^2 \epsilon^2} + |\mathcal{S}||\mathcal{A}|,$$

$$D_2 = \frac{\overline{C}_2 \log\left(4|\mathcal{S}|/\delta\right)}{(1-\gamma)^2 (1-\gamma^K)^2 \alpha^2 \epsilon^2},$$

*where $\overline{C}_1$, $\overline{C}_2$ are absolute constants, then with probability at least $1 - \delta$, the learned Bayesian value function satisfies*

$$\max_s |\widehat{V}_{K,\mathcal{A}^-,\varepsilon}^{\mathrm{Bayes},*}(s) - V_{K,\mathcal{A}^-,\varepsilon}^{\mathrm{Bayes},*}(s)| \leq \epsilon. \tag{13}$$

Theorem 5.1 quantifies the sample complexity of BOLA, explicitly capturing the interplay between environment sampling and predictive adaptation. In particular, BOLAs total sample requirements decompose into two distinct regimes:

**Environmentinteraction samples $D_1$.** The first requirement $D_1$, arising from direct environment interactions, scales with $|\mathcal{A}| - |\mathcal{A}^-|$, the number of unpredictable actions. This leads to a sample complexity which is strictly smaller than the classical dependence of $O(|\mathcal{S}||\mathcal{A}|\epsilon^{-2})$ [14, 24], since increased predictability reduces the environment sampling burden. In the extreme case where all actions in the prediction horizon are predictable ($\mathcal{A}^- = \mathcal{A}$), the dominant term of $D_1$ vanishes altogether (except for the reward function learning cost $|\mathcal{S}||\mathcal{A}|$), as the predictive model fully specifies the transition dynamics within the prediction horizon.

**Prediction Oracle Samples $D_2$.** The second term $D_2$ represents the samples required from the predictive model, which exhibits a distinct scaling behavior: as the prediction horizon $K$ grows, the required number of samples decreases due to the stronger contraction factor $\gamma^K$ in the Bayesian Bellman equation. When $K \geq \mathcal{O}(\log(\frac{1}{\gamma}))$, this term improves to $(1 - \gamma)^{-2}$, which is lower than that in model-based RL [15]. This highlights that when predictions are both comprehensive and with long enough horizon, BOLA can achieve lower sample complexity than classical MDP approaches.

**Trade-off between the Two Sample Sources.** Together, these two sampling regimes reveal a trade-off parameterized by $\alpha$: increasing $\alpha$ (more environment interaction) raises the environment sample requirement $D_1$ to $O((1 - \alpha)^{-2})$, while reducing the required number of samples $D_2$ from the prediction oracle. One can choose a reasonable $\alpha$ to trade off the sample requirements.

## 6 Numerical Studies

Although our emphasis is on theory and finitesample guarantees, we provide a small-scale empirical case study to demonstrate practical relevance. Specifically, we evaluate BOLA on a windfarm storage control task, where the operator minimizes energy imbalance penalties by charging or discharging a battery based on wind mismatch, price signals, and current state of charge (see Figure 1 for the setup, Appendix G for more details, and Appendix H for additional experiments).

Specifically, Figure 2(a) illustrates the cumulative cost reduction achieved by BOLA under different prediction horizons, compared against a baseline MDP policy without prediction. As the prediction horizon $K$ increases from 1 to 4, the cost reduction consistently improves, confirming that longer foresight enables the agent to better anticipate upcoming mismatches and price fluctuations. Notably, the largest marginal improvement occurs at $K = 1$, indicating that even a short look-ahead can significantly enhance decision-making performance. Figure 2(b) illustrates the robustness of BOLA under increasing levels of relative prediction error. As the prediction error increases, the performance of all methods declines approximately linearly, which aligns with our theoretical analysis. Notably, longer prediction horizons yield greater cost savings under perfect or low-noise forecasts but also exhibit greater sensitivity to prediction errors. In contrast, shorter horizons are more stable and

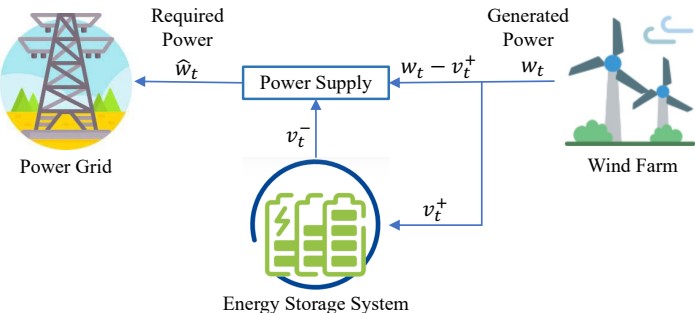

Figure 1: Wind Farm Storage Control

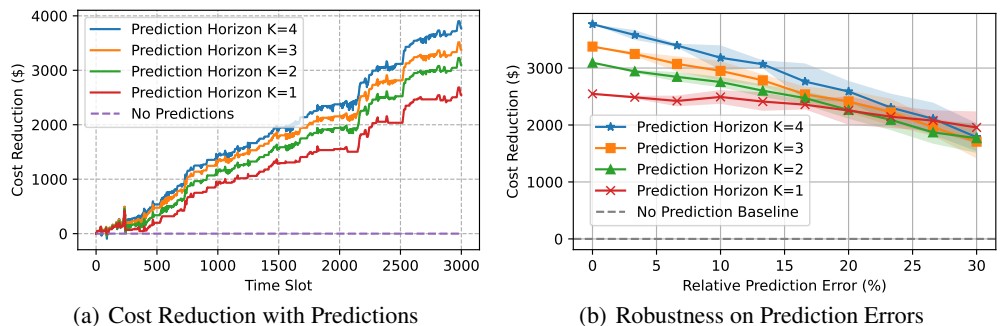

(a) Cost Reduction with Predictions      (b) Robustness on Prediction Errors

Figure 2: Storage Control Performance. (a) Cumulative cost reduction for different prediction horizons: longer prediction horizon yields greater savings over the no-prediction baseline. (b) Robustness to prediction noise: cost savings decline roughly linearly with error, yet all predictive policies outperform the baseline even at 30% noise.

degrade more gradually as noise increases. Nevertheless, all prediction-based policies consistently outperform the no-prediction baseline, even when the relative error reaches 30%.

## 7   Conclusion

In this work, we study the theoretical value of transition predictions in sequential decision-making. We propose a prediction-augmented MDP framework, characterize the benefit of predictions via the Bellman-Jensen Gap, and develop a tractable model-based RL algorithm with sample complexity guarantees. A natural future direction is to extend our results to the model-free setting.

**Limitations and Future Directions.** In prediction-augmented MDP, we consider fixed-horizon planning where the agent receives a $K$-step prediction and plans a $K$-step sequence of actions. For sequential decision-making problems with predictions, another popular framework is called the receding-horizon control, where the agent receives a $K$-step prediction but only plans a single-step action instead of a sequence of $K$ actions. Intuitively, using receding-horizon control could be more beneficial to the agent than fixed-horizon planning, since the agent does not have to commit to a sequence of actions and can adaptively choose actions based on the new realizations of the states and the predictions. Further investigating the advantage of prediction-augmented MDPs with receding-horizon control is among the future directions of this work. On the theoretical side, we have established the first upper bounds on BOLAs sample complexity, but it remains open whether these can be tightened or matched by lower bounds. In particular, refined variance-based techniques (e.g., refined concentration for the multi-step Bayesian operator) may yield stronger guarantees.

## Acknowledgments

We sincerely thank the anonymous area chair and reviewers for their insightful feedback. We are grateful to Hongyu Yi for proofreading the paper and to Laixi Shi for helpful discussions.

C. Wu's work was supported in part by the National Natural Science Foundation of China under Grant 72271213, the Shenzhen Science and Technology Program under Grant RCYX20221008092927070. A. Wierman's work was supported in part by NSF grants CNS-2146814, CPS-2136197, CNS-2106403, and NGSDI-2105648.

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

# Appendices

## A  Related Literature

**Classical Model-Based RL and Sample Complexity.** Our work builds on the modelbased RL paradigm, which separates learning into model estimation and policy planning. Classical model-based RL for finite MDPs has been extensively studied and achieves minimax-optimal sample complexity guarantees [14, 25–29]. Recent extensions of sample complexity analyses cover broader classes of problems, including average-reward MDPs [30], constrained MDPs [31], and partially observable MDPs (POMDPs) [32–34]. In contrast, we tackle a new settingMDPs augmented with multistep transition predictionsand establish the first finitesample guarantees (Theorem 5.1) under both environment and prediction sampling. Notably, when the prediction horizon satisfies $K = \Omega(\log(1/\gamma))$, the stronger contraction factor $\gamma^K$ leads to a strictly lower sample complexity than the classical bound [14, 15], highlighting how predictive information can fundamentally accelerate learning.

**RL with Look-ahead Prediction and Planning**. Integrating multi-step look-ahead forecasts into reinforcement learning has driven strong empirical gains. Related work ranges from Bayes-Adaptive methods [35–37] and look-ahead Q-learning [38, 39] to policy-iteration variants [40, 41] and real-time dynamic programming [42]. However, these methods focus primarily on empirical gains and offer little theoretical support on some key points: they neither quantify the value added by multi-step predictions nor guarantee robustness when forecasts are noisy or partial, and they do not address the sample complexity of learning with such predictions. We fill these gaps by introducing the BellmanJensen framework, which is the first to deliver rigorous performance and robustness guarantees for multi-step, imperfect transition predictions in RL. We also propose BOLA, which comes with finite-sample complexity bounds for RL with transition predictions.

**MDPs with Predictions.** Recent work has explored integrating predictions into MDPs in several ways. Some methods leverage $Q$-value predictions to to speed up learning or refine suboptimal policies [43, 44]. There are also some more related work focusing on incorporating estimates of the transition kernel to boost performance in nonstationary environment [17] or improve sample efficiency [18]. However, these approaches target matching the performance of an idealized MDP with perfect models. By contrast, we use realizationbased, multistep transition forecasts to surpass the inherent limits of classical online MDP solutions, driving performance closer to the offlineoptimal benchmark even under imperfect and partial predictions (Theorem 4.1).

**Online Optimization and Control with Predictions.** Leveraging predictive information is a common strategy in broader online optimization and control contexts, notably within Online Convex Optimization (OCO) and Model Predictive Control (MPC). For example, recent studies [45–47] demonstrate exponentially decaying regret when leveraging predictions in Smoothed Online Convex Optimization (SOCO). Lin *et al.* [48] have identified conditions under which predictions significantly enhance performance in general online optimization settings. Similarly, prediction-driven improvements have been established in linear-quadratic control frameworks, yielding exponential regret reduction [49, 50], and have been successfully extended to MPC settings, even with time-varying constraints [51, 52]. Mercier *et al.* [53] investigate prediction in a general online optimization setting, offering useful insights, while leaving open questions on the formal analysis on the value of prediction and on broader prediction settings. Compared to this extensive literature on deterministic or structured linear settings, prediction-augmented stochastic models such as MDPs have received relatively limited attention due to their inherent complexity and lack of closed-form solutions. Our work addresses this gap by providing rigorous theoretical foundations, clear sample complexity characterizations, and practical algorithms that effectively integrate predictive realizations into sequential stochastic decision-making.

# B Proof of Theorem 3.1

The proof is divided into 3 steps. First, we construct an auxiliary MDP $\tilde{M}$ and show that its optimal policy $\tilde{\pi}^*$ is the same as the optimal policy $\pi^*$ of the prediction-augmented MDP. Then, we show that the optimal value function satisfies our proposed Bellman equation of the prediction-augmented MDP. Finally, we show that the Bellman operator associated with our Bellman equation is a contraction mapping, which implies that it has a unique fixed point solution.

**Step 1: Construction of the Auxiliary MDP $\tilde{M}$.** In a prediction-augmented MDP, the agent essentially makes decisions based on both the current state $s$ and the received prediction $\boldsymbol{\sigma}$. Therefore, we can incorporate the prediction into the state and define an auxiliary MDP based on it. Specifically, let $\tilde{M} = (\tilde{\mathcal{S}}, \tilde{\mathcal{A}}, \tilde{r}, \tilde{P}, \tilde{\gamma})$ be an MDP with state space $\tilde{\mathcal{S}}$, action space $\tilde{\mathcal{A}}$, reward function $\tilde{r}$, transition kernel $\tilde{P}$, and discount factor $\gamma$. The state space $\tilde{\mathcal{S}}$ is the product space of the state space of the original MDP and the domain of the prediction $\boldsymbol{\sigma}$, i.e., $\tilde{s} = (s, \boldsymbol{\sigma}) \in \tilde{\mathcal{S}} := \mathcal{S} \times \mathcal{Q}$, where $\boldsymbol{\sigma} = (\sigma_1, \sigma_2, ..., \sigma_K)$. The action space $\tilde{\mathcal{A}}$ is the $K$-product space of the action space of the original MDP, i.e., $\boldsymbol{a} = (a_0, a_1, \cdots, a_{K-1}) \in \tilde{\mathcal{A}} := \mathcal{A}^K$. For any $\tilde{s} \in \tilde{\mathcal{S}}$ and $\boldsymbol{a} \in \tilde{\mathcal{A}}$, the reward function $\tilde{r}(\cdot, \cdot)$ is defined to be the $K$-step expected discounted reward of the original MDP, that is,

$$\tilde{r}(\tilde{s}, \boldsymbol{a}) = \mathbb{E}\left[\sum\nolimits_{t=0}^{K-1} \gamma^t r(s_t, a_t)\right],$$

where $s_0 = s$ and $s_t \sim P(\cdot \mid s, \boldsymbol{a}_{0:t-1}, \boldsymbol{\sigma}_{1:t})$ for all $t \geq 1$. For any $(\tilde{s} = (s, \boldsymbol{\sigma}), \boldsymbol{a})$ and $\tilde{s}' = (s', \boldsymbol{\sigma}')$, the transition probability $\tilde{P}(\tilde{s}'|\tilde{s}, \boldsymbol{a})$ is defined as

$$\tilde{P}(\tilde{s}'|\tilde{s}, \boldsymbol{a}) = P(s', \boldsymbol{\sigma}'|s, \boldsymbol{\sigma}, \boldsymbol{a}) = P(s'|s, \boldsymbol{a}, \boldsymbol{\sigma})P(\boldsymbol{\sigma}').$$

Finally, the discount factor of the auxiliary MDP satisfies $\tilde{\gamma} = \gamma^K$.

For the policy $\pi : \tilde{\mathcal{S}} \times \tilde{\mathcal{A}} \to \Delta(\tilde{\mathcal{A}})$ (where $\Delta(\tilde{\mathcal{A}})$ denotes the probability simplex on $\tilde{\mathcal{A}}$). The corresponding value function $\tilde{V}^\pi$ of the auxiliary MDP is defined as

$$\tilde{V}^\pi(\tilde{s}) = \mathbb{E}_\pi\left[\sum\nolimits_{t=0}^\infty \gamma^t \tilde{r}(\tilde{s}_t, \boldsymbol{a}_t)\middle| \tilde{s}_0 = \tilde{s}\right], \quad \forall \tilde{s} \in \tilde{\mathcal{S}}.$$

The auxiliary MDP is with an infinite state space and a finite action space. The Bellman optimality equation remains valid provided the bounded rewards, the measurable transition kernel, and a discount factor $\tilde{\gamma} \in [0, 1)$. These conditions ensure the Bellman operator is a $\tilde{\gamma}$-contraction on the space of bounded measurable functions [54, 55].

Therefore, the optimal value function $\tilde{V}^*$ is the unique solution to the following Bellman optimality equation:

$$\tilde{V}^*(\tilde{s}) = \max_{\boldsymbol{a} \in \tilde{\mathcal{A}}} \left(\tilde{r}(\tilde{s}, \boldsymbol{a}) + \gamma^K \int_{\tilde{s} \in \tilde{\mathcal{S}}} \tilde{P}(d\tilde{s}'|\tilde{s}, \boldsymbol{a})\tilde{V}^*(\tilde{s}')\right). \tag{14}$$

In addition, any policy $\tilde{\pi}^*$ satisfying

$$\{\boldsymbol{a} \in \tilde{\mathcal{A}} \mid \tilde{\pi}^*(\boldsymbol{a} \mid \tilde{s}) > 0\} \subseteq \arg\max_{\boldsymbol{a} \in \tilde{\mathcal{A}}} \left(\tilde{r}(\tilde{s}, \boldsymbol{a}) + \gamma^K \int_{\tilde{s} \in \tilde{\mathcal{S}}} \tilde{P}(d\tilde{s}'|\tilde{s}, \boldsymbol{a})\tilde{V}^*(\tilde{s}')\right) \tag{15}$$

for all $\tilde{s} = (s, \boldsymbol{\sigma})$ is an optimal policy.

Since the auxiliary MDP has the same problem structure (state, action, transition, reward) as the prediction-augmented MDP, an optimal policy $\tilde{\pi}^*$ of the auxiliary MDP is also an optimal policy $\pi^*$ of the prediction-augmented MDP.

**Step 2: Establishing the Bellman Equation.** Recall from Section 3.1 that we defined the Bayesian value function $V^{\text{Bayes},\pi}(s) = \mathbb{E}_{\boldsymbol{\sigma}}[V_{K,\mathcal{A}^-,\varepsilon}^\pi(s, \boldsymbol{\sigma})] = \mathbb{E}_{\boldsymbol{\sigma}}[\tilde{V}^\pi(\tilde{s})]$, where $\tilde{s} = (s, \boldsymbol{\sigma})$. Using the previous identity in Eq. (14), we have that under the optimal policy $\pi^*$,

$$V_{K,\mathcal{A}^-,\varepsilon}^{\text{Bayes},*}(s) = \mathbb{E}_{\boldsymbol{\sigma}}\left[\max_{\boldsymbol{a} \in \tilde{\mathcal{A}}} \left(\tilde{r}(\tilde{s}, \boldsymbol{a}) + \gamma^K \int_{\tilde{s} \in \tilde{\mathcal{S}}} \tilde{P}(d\tilde{s}'|\tilde{s}, \boldsymbol{a})\tilde{V}^*(\tilde{s}')\right)\right]$$

$$= \mathbb{E}_{\boldsymbol{\sigma}}\left[\max_{\boldsymbol{a} \in \tilde{\mathcal{A}}} \left[\sum\nolimits_{t=0}^{K-1} \gamma^t \left(\sum\nolimits_{s_t} P(s_t|s, \boldsymbol{a}_{0:t-1}, \boldsymbol{\sigma}_{1:t})r(s_t, a_t)\right)\right.\right.$$

$$+\gamma^K \int_{\tilde{s}\in\tilde{S}} \tilde{P}(d\tilde{s}'|\tilde{s},\boldsymbol{a})\tilde{V}^*(\tilde{s}')\Bigg]. \tag{16}$$

For the term $\int_{\tilde{s}\in\tilde{S}} \tilde{P}(d\tilde{s}'|\tilde{s},\boldsymbol{a})\tilde{V}^*(\tilde{s}')$, we use the independence between prediction $\boldsymbol{\sigma}$ and state $s$ and get:

$$\begin{aligned}
\int_{\tilde{s}\in\tilde{S}} \tilde{P}(d\tilde{s}'|\tilde{s},\boldsymbol{a})\tilde{V}^*(\tilde{s}') &= \sum_{s'\in S} \int_{\boldsymbol{\sigma}'\in\mathcal{Q}} P(s'|s,\boldsymbol{a},\boldsymbol{\sigma})P(d\boldsymbol{\sigma}')\tilde{V}^*(s',\boldsymbol{\sigma}') \\
&= \sum_{s'\in S} P(s'|s,\boldsymbol{a},\boldsymbol{\sigma}) \int_{\boldsymbol{\sigma}'\in\mathcal{Q}} P(d\boldsymbol{\sigma}')\tilde{V}^*(s',\boldsymbol{\sigma}') \\
&= \sum_{s'\in S} P(s'|s,\boldsymbol{a},\boldsymbol{\sigma})\mathbb{E}_{\boldsymbol{\sigma}'}[\tilde{V}^*(s',\boldsymbol{\sigma}')] \\
&= \sum_{s'\in S} P(s'|s,\boldsymbol{a},\boldsymbol{\sigma})\mathbb{E}_{\boldsymbol{\sigma}'}[V^*_{K,\mathcal{A}^-,\varepsilon}(s',\boldsymbol{\sigma}')] \\
&= \sum_{s'\in S} P(s'|s,\boldsymbol{a},\boldsymbol{\sigma})V^{\text{Bayes},*}_{K,\mathcal{A}^-,\varepsilon}(s'). \tag{17}
\end{aligned}$$

Combining the previous two equations, we finally have:

$$\begin{aligned}
V^{\text{Bayes},*}_{K,\mathcal{A}^-,\varepsilon}(s) = \mathbb{E}_{\boldsymbol{\sigma}}\Bigg[\max_{\boldsymbol{a}\in\tilde{\mathcal{A}}}\Bigg[&\sum_{t=0}^{K-1}\gamma^t\Big(\sum_{s_t}P(s_t|s,\boldsymbol{a}_{0:t-1},\boldsymbol{\sigma}_{1:t})r(s_t,a_t)\Big) \\
&+\gamma^K\sum_{s'\in S}P(s'|s,\boldsymbol{a},\boldsymbol{\sigma})V^{\text{Bayes},*}_{K,\mathcal{A}^-,\varepsilon}(s')\Bigg]. \tag{18}
\end{aligned}$$

for all $s \in \mathcal{S}$. This leads to the Bellman optimality equation in Eq. (5).

**Step 3: The Uniqueness of the Solution to the Bayesian Bellman Equation.** It is easy to verify that our Bellman equation is a fixed-point equation with a contractive fixed-point operator, where the contraction factor is $\gamma^K$. Therefore, by the Banach fixed-point theorem [56], the solution to our Bellman equation is unique.

## C    Proof of the Corollary 3.1

This is a direct corollary of Theorem 3.1. Recall that, the optimal policy $\pi^*$ is consistent with the optimal policy of the auxiliary MDP $\tilde{M}$ defined in Section B. Hence, combining Eq. (15) and Eq. (17) yields the optimal policy:

$$\begin{aligned}
\{\boldsymbol{a}\in\mathcal{A}^K \mid \pi^*(\boldsymbol{a}\mid s,\boldsymbol{\sigma}) > 0\} \subseteq \arg\max_{\boldsymbol{a}\in\mathcal{A}^K}\Bigg(&\sum_{t=0}^{K-1}\gamma^t\Big(\sum_{s_t}P(s_t|s,\boldsymbol{a}_{0:t-1},\boldsymbol{\sigma}_{1:t})r(s_t,a_t)\Big) \\
&+\gamma^K\sum_{s_K}P(s_K|s,\boldsymbol{a},\boldsymbol{\sigma})V^{\text{Bayes},*}_{K,\mathcal{A}^-,\varepsilon}(s_K)\Bigg), \forall s\in\mathcal{S},\boldsymbol{\sigma}\in\mathcal{Q},
\end{aligned}$$

## D    Proof of the Existence of $V^{\text{Bayes},*}_{\text{off}}(s)$

For any state $s\in\mathcal{S}$ and $k\geq 1$, we define the following truncated optimal value function with $k$-step accurate transition prediction $\boldsymbol{\sigma}^*_{1:k}$:

$$V^*_{\text{off},k}(s) = \mathbb{E}_{\boldsymbol{\sigma}^*_{1:k}}\left(\max_{\boldsymbol{a}_{0:k-1}}\left(\sum_{t=0}^{k-1}\gamma^t r(s_t,a_t|s_0 = s,\boldsymbol{\sigma}^*_{1:t+1})\right)\right).$$

We now show the sequence $\{V^*_{\text{off},k}(s)\}_k$ is (1) monotonically increasing with $k$ and (2) bounded.

For monotonically increasing, since $V^*_{\text{off},k+1}(s)$ can be represented by:

$$\begin{aligned}
V^*_{\text{off},k+1}(s) &= \mathbb{E}_{\boldsymbol{\sigma}^*_{1:k+1}}\left(\max_{\boldsymbol{a}_{0:k}}\left(\sum_{t=0}^{k}\gamma^t r(s_t,a_t|s_0 = s,\boldsymbol{\sigma}^*_{1:t+1})\right)\right) \\
&\geq \mathbb{E}_{\boldsymbol{\sigma}^*_{1:k+1}}\left(\max_{\boldsymbol{a}_{0:k-1}}\left(\sum_{t=0}^{k-1}\gamma^t r(s_t,a_t|s_0 = s,\boldsymbol{\sigma}^*_{1:t+1})\right)+\min_{a_k}\gamma^k r(s_k,a_k|s_0 = s,\sigma^*_{k+1})\right)
\end{aligned}$$

$$\geq \mathbb{E}_{\boldsymbol{\sigma}^*_{1:k+1}} \left( \max_{\boldsymbol{a}_{0:k-1}} \left( \sum_{t=0}^{k-1} \gamma^t r(s_t, a_t | s_0 = s, \boldsymbol{\sigma}^*_{1:t+1}) \right) \right)$$

$$= V^*_{\text{off},k}(s),$$

where the last inequality is due to $r(s,a) \geq 0$ for any state-action pair $(s,a)$.

For the bounded property, it is clear that $V^*_{\text{off},k}(s) \leq \sum_{t=0}^{k-1} \gamma^t \leq \frac{1}{1-\gamma}$ for any $k$. Hence, the sequence $\{V^*_{\text{off},k}(s)\}_k$ is monotonically increasing and bounded, implying $V^{\text{Bayes},*}_{\text{off}}(s) = \lim_{k \to \infty} V^*_{\text{off},k}(s)$ exists for any $s$. This concludes our proof.

## E Proof of Theorem 4.1

Let $V^{\text{Bayes},*}_{\infty,\mathcal{A}^-,\boldsymbol{0}}(s) := \lim_{k \to \infty} V^{\text{Bayes},*}_{k,\mathcal{A}^-,\boldsymbol{0}}(s)$ for all $s \in \mathcal{S}$. The following lemma verifies that $V^{\text{Bayes},*}_{\infty,\mathcal{A}^-,\boldsymbol{0}}(s)$ is indeed well-defined.

**Lemma E.1** (Proof in Appendix E.1). $\lim_{k \to \infty} V^{\text{Bayes},*}_{k,\mathcal{A}^-,\boldsymbol{0}}(s)$ *exists and is unique.*

To bound the difference between $V^{\text{Bayes},*}_{\text{off}}(s)$ and $V^{\text{Bayes},*}_{K,\mathcal{A}^-,\boldsymbol{\varepsilon}}(s)$, we perform the following decomposition:

$$V^{\text{Bayes},*}_{\text{off}}(s) - V^{\text{Bayes},*}_{K,\mathcal{A}^-,\boldsymbol{\varepsilon}}(s) = \underbrace{V^{\text{Bayes},*}_{\infty,\mathcal{A}^-,\boldsymbol{0}}(s) - V^{\text{Bayes},*}_{K,\mathcal{A}^-,\boldsymbol{0}}(s)}_{T_1} + \underbrace{V^{\text{Bayes},*}_{\text{off}}(s) - V^{\text{Bayes},*}_{\infty,\mathcal{A}^-,\boldsymbol{0}}(s)}_{T_2}$$

$$+ \underbrace{V^{\text{Bayes},*}_{K,\mathcal{A}^-,\boldsymbol{0}}(s) - V^{\text{Bayes},*}_{K,\mathcal{A}^-,\boldsymbol{\varepsilon}}(s)}_{T_3}, \forall s \in \mathcal{S}. \tag{19}$$

Here, $T_1$ denotes the loss incurred by the finite prediction window $K$, $T_2$ captures the loss stemming from partial action predictability, and $T_3$ accounts for the error introduced by prediction errors. We bound these three terms as follows.

**Step 1: Bounding the Term $T_1$ using Dyadic Horizon Decomposition.** We begin by analyzing the Bellman-Jensen Gap due to the finite prediction horizon $K$. Since the value function sequence $\{V^{\text{Bayes},*}_{K,\mathcal{A}^-,\boldsymbol{0}}(s)\}$ converges as $k \to \infty$, any subsequence must also converge to the same limit. Therefore, for any $s \in \mathcal{S}$, we make a dyadic horizon decomposition as follows:

$$V^{\text{Bayes},*}_{\infty,\mathcal{A}^-,\boldsymbol{0}}(s) - V^{\text{Bayes},*}_{K,\mathcal{A}^-,*}(s) = \lim_{k \to \infty} \left( V^{\text{Bayes},*}_{k,\mathcal{A}^-,\boldsymbol{0}}(s) - V^{\text{Bayes},*}_{K,\mathcal{A}^-,\boldsymbol{0}}(s) \right)$$

$$= \lim_{k \to \infty} \left( V^{\text{Bayes},*}_{K \cdot 2^k,\mathcal{A}^-,\boldsymbol{0}}(s) - V^{\text{Bayes},*}_{K,\mathcal{A}^-,\boldsymbol{0}}(s) \right)$$

$$= \lim_{k \to \infty} \left( \sum_{i=0}^{k-1} \left( V^{\text{Bayes},*}_{K \cdot 2^{i+1},\mathcal{A}^-,\boldsymbol{0}}(s) - V^{\text{Bayes},*}_{K \cdot 2^i,\mathcal{A}^-,\boldsymbol{0}}(s) \right) \right). \tag{20}$$

Based on this decomposition, we only need to provide an upper bound to the value function gap when doubling the prediction window, which is stated in the following lemma.

**Lemma E.2** (Proof in Appendix E.2). *For any prediction window $K \geq 1$, we have*

$$\max_{s \in \mathcal{S}} \left( V^{\text{Bayes},*}_{2K,\mathcal{A}^-,\boldsymbol{0}}(s) - V^{\text{Bayes},*}_{K,\mathcal{A}^-,\boldsymbol{0}}(s) \right) \leq \frac{\gamma^K \sqrt{CK \log |\mathcal{A}|}}{(1-\gamma)(1-\gamma^{2K})}, \tag{21}$$

*where $C$ is an absolute constant.*

By repeatedly using Lemma E.2, we have the following lemma that bounds the term $T_1$ in Eq. (19).

**Lemma E.3** (Proof in Appendix E.3). *There exists an absolute constant $C_0 > 0$ such that the following inequality holds for all $K \geq C_0$:*

$$T_1 \leq \frac{C_1 \gamma^K \sqrt{K \log |\mathcal{A}|}}{(1-\gamma)^{\frac{6}{5}}(1-\gamma^{2K})},$$

*where $C_1$ is an absolute constant.*

**Step 2: Bounding the term $T_2$.** For any $s \in \mathcal{S}$, $V_{\text{off}}^{\text{Bayes},*}(s)$ and $V_{\infty,\mathcal{A}^-,\mathbf{0}}^{\text{Bayes},*}(s)$ are based on full prediction $\boldsymbol{\sigma}^*$ and partial one $\boldsymbol{\sigma}$ with predictable action $a \in \mathcal{A}^-$, satisfying:

$$V_{\text{off}}^{\text{Bayes},*}(s) = \lim_{k \to \infty} \mathbb{E}_{\boldsymbol{\sigma}_{1:k}^*} \left( \max_{\boldsymbol{a}_{0:k-1}} \left( \sum_{t=0}^{k-1} \gamma^t r(s_t, a_t | s_0 = s, \boldsymbol{\sigma}_{1:t+1}^*) \right) \right),$$

$$V_{\infty,\mathcal{A}^-,\mathbf{0}}^{\text{Bayes},*}(s) = \lim_{k \to \infty} \mathbb{E}_{\boldsymbol{\sigma}_{1:k}^*} \left( \max_{\boldsymbol{a}_{0:k-1}} \left( \sum_{t=0}^{k-1} \gamma^t r(s_t, a_t | s_0 = s, \boldsymbol{\sigma}_{1:t+1}) \right) \right).$$

Therefore, we have

$$V_{\text{off}}^{\text{Bayes},*}(s) - V_{\infty,\mathcal{A}^-,\mathbf{0}}^{\text{Bayes},*}(s) = \lim_{k \to \infty} \mathbb{E}_{\boldsymbol{\sigma}_{1:k}^*} \left( \max_{\boldsymbol{a}_{0:k-1}} \left( \sum_{t=0}^{k-1} \gamma^t r(s_t, a_t | s_0 = s, \boldsymbol{\sigma}_{1:t+1}^*) \right. \right.$$
$$\left. \left. - \sum_{t=0}^{k-1} \gamma^t r(s_t, a_t | s_0 = s, \boldsymbol{\sigma}_{1:t+1}) \right) \right).$$

The following lemma further bounds the Bellman-Jensen Gap due to partial action-coverage.

**Lemma E.4** (Proof in Appendix E.4). *For any predictable action set $\mathcal{A}^- \subseteq \mathcal{A}$, we have:*

$$\max_s \left( V_{\text{off}}^{\text{Bayes},*}(s) - V_{\infty,\mathcal{A}^-,\mathbf{0}}^{\text{Bayes},*}(s) \right) \le C_2 \sum_{t=1}^{\infty} \gamma^t \sqrt{\log(|\mathcal{A}|^{t+1} - |\mathcal{A}^-|^{t+1} + 1)\theta_{\max}^2},$$

*where $C_2$ is an absolute constant, $\theta_{\max}^2 = \max_{s,\boldsymbol{a}_{0:t},t} \sigma^2(r(s_t, a_t | s_0 = s, \boldsymbol{a}_{0:t}))$, where $\sigma(\cdot)$ denotes the sub-Gaussian parameter. Parameter $\theta_{\max}^2$ then captures the variability of the reward.*

**Step 3: Bounding the Term $T_3$.** $T_3$ captures the value decay due to prediction errors. We

**Lemma E.5** (Proof in Appendix E.5). *For any prediction horizon $K$ and predictable action set $\mathcal{A}^-$, we have:*

$$\max_{s \in \mathcal{S}}(V_{K,\mathcal{A}^-,\mathbf{0}}^{\text{Bayes},*}(s) - V_{K,\mathcal{A}^-,\boldsymbol{\varepsilon}}^{\text{Bayes},*}(s)) \le \sum_{j=1}^{K} \frac{\gamma^j}{(1-\gamma)(1-\gamma^K)} \epsilon_j, \tag{22}$$

*where $\epsilon_j := W_1^{(d)}\left(\mathcal{P}_j^*, \widehat{\mathcal{P}}_j\right)$ denotes the Wasserstein1 distance between the distributions $\mathcal{P}_j^*$, $\widehat{\mathcal{P}}_j$ of the $j$-step predictive model under accurate and inaccurate prediction $\sigma_j^*$ and $\hat{\sigma}_j$, respectively, measured with respect to the base metric $d(\sigma_j, \sigma_j') = W_1(\sigma_j, \sigma_j')$, i.e., the Wasserstein1 distance between single-step predictive distributions.*

**Step 4: Putting pieces together.** Our last step is to combine the bounds we obtained for the terms $T_1$, $T_2$ and $T_3$ to get the final results:

$$\max_s \left( V_{\text{off}}^*(s) - V_{+K}^*(s) \right) \le \frac{C_1 \gamma^K \sqrt{K \log |\mathcal{A}|}}{(1-\gamma)^{\frac{6}{5}}(1-\gamma^{2K})} + \sum_{j=1}^{K} \frac{\gamma^j}{(1-\gamma)(1-\gamma^K)} \epsilon_j$$
$$+ C_2 \sum_{t=1}^{\infty} \gamma^t \sqrt{\log(|\mathcal{A}|^{t+1} - |\mathcal{A}^-|^{t+1} + 1)\theta_{\max}^2}.$$

This concludes the proof.

### E.1 Proof of Lemma E.1

First of all, since we work with bounded rewards, it is easy to see that $V_{k,\mathcal{A}^-,\mathbf{0}}^{\text{Bayes},*}(s) \le 1/(1-\gamma)$ for any $k \ge 0$ and $s \in \mathcal{S}$. Next, we show that $\{V_{k,\mathcal{A}^-,\mathbf{0}}^{\text{Bayes},*}(s)\}_{k \ge 1}$ is a Cauchy sequence, which implies its convergence [57].

Recall the Bellman equation for $V_{k+1,\mathcal{A}^-,\mathbf{0}}^{\text{Bayes},*}$:

$$V_{k+1,\mathcal{A}^-,\mathbf{0}}^{\text{Bayes},*}(s) = \mathbb{E}_{\boldsymbol{\sigma}_{1:k+1}} \left[ \max_{\boldsymbol{a}} \left( \sum_{t=0}^{k} \gamma^t \left( \sum_{s_t} P(s_t|s, \boldsymbol{a}_{0:t-1}, \boldsymbol{\sigma}_{1:t}) r(s_t, a_t) \right) \right. \right.$$
$$\left. \left. + \gamma^{k+1} \sum_{s_{k+1}} P(s_{k+1}|s, \boldsymbol{a}, \boldsymbol{\sigma}) V_{k+1,\mathcal{A}^-,\mathbf{0}}^{\text{Bayes},*}(s_{k+1}) \right) \right]$$

for any $s \in \mathcal{S}$ and $k \geq 0$. Therefore, we can truncate the tail term and have

$$V_{k+1,\mathcal{A}^-,\mathbf{0}}^{\text{Bayes},*}(s) \leq \underbrace{\mathbb{E}_{\boldsymbol{\sigma}_{1:k}}\left[\max_{\boldsymbol{a}}\left(\sum_{t=0}^{k}\gamma^t\left(\sum_{s_t}P(s_t|s,\boldsymbol{a}_{0:t-1},\boldsymbol{\sigma}_{1:t})r(s_t,a_t)\right)\right)\right]}_{:=\overline{V}_{k,\mathcal{A}^-,\mathbf{0}}^{\text{Bayes},*}(s)} + \frac{\gamma^{k+1}}{1-\gamma}.$$

In addition, since the residual is positive, we have $V_{k+1,\mathcal{A}^-,\mathbf{0}}^{\text{Bayes},*}(s) - \overline{V}_{k,\mathcal{A}^-,\mathbf{0}}^{\text{Bayes},*}(s) \geq 0$. Together, they imply

$$\left|V_{k+1,\mathcal{A}^-,\mathbf{0}}^{\text{Bayes},*}(s) - \overline{V}_{k,\mathcal{A}^-,\mathbf{0}}^{\text{Bayes},*}(s)\right| \leq \frac{\gamma^k}{1-\gamma}. \tag{23}$$

Similarly, we have

$$V_{k,\mathcal{A}^-,\mathbf{0}}^{\text{Bayes},*}(s)$$

$$= \mathbb{E}_{\boldsymbol{\sigma}}\left[\max_{\boldsymbol{a}}\left(\sum_{t=0}^{k-1}\gamma^t\left(\sum_{s_t}P(s_t|s,\boldsymbol{a}_{0:t-1},\boldsymbol{\sigma}_{1:t})r(s_t,a_t)\right) + \gamma^k\sum_{s_K}P(s_K|s,\boldsymbol{a},\boldsymbol{\sigma})V_{+k}^*(s_K)\right)\right]$$

$$\leq \mathbb{E}_{\boldsymbol{\sigma}}\left[\max_{\boldsymbol{a}}\left(\sum_{t=0}^{k}\gamma^t\sum_{s_t}P(s_t|s,\boldsymbol{a}_{0:t},\boldsymbol{\sigma}_{1:t+1})r(s_t,a_t)\right)\right] + \frac{\gamma^k}{1-\gamma}$$

$$= \overline{V}_{k,\mathcal{A}^-,\mathbf{0}}^{\text{Bayes},*}(s) + \frac{\gamma^k}{1-\gamma}$$

and $V_{k,\mathcal{A}^-,\mathbf{0}}^{\text{Bayes},*}(s) - \overline{V}_{k,\mathcal{A}^-,\mathbf{0}}^{\text{Bayes},*}(s) \geq -\gamma^k$. Together, they imply

$$\left|V_{k,\mathcal{A}^-,\mathbf{0}}^{\text{Bayes},*}(s) - \overline{V}_{k,\mathcal{A}^-,\mathbf{0}}^{\text{Bayes},*}(s)\right| \leq \frac{\gamma^k}{1-\gamma}, \forall s \in \mathcal{S}. \tag{24}$$

Combining Eq. (23) and (24), we have by triangle inequality that

$$\left|V_{k+1,\mathcal{A}^-,\mathbf{0}}^{\text{Bayes},*}(s) - V_{k,\mathcal{A}^-,\mathbf{0}}^{\text{Bayes},*}(s)\right| \leq \left|V_{k+1,\mathcal{A}^-,\mathbf{0}}^{\text{Bayes},*}(s) - \overline{V}_{k,\mathcal{A}^-,\mathbf{0}}^{\text{Bayes},*}(s)\right| + \left|\overline{V}_{k,\mathcal{A}^-,\mathbf{0}}^{\text{Bayes},*}(s) - V_{k,\mathcal{A}^-,\mathbf{0}}^{\text{Bayes},*}(s)\right|$$

$$\leq \frac{2\gamma^k}{1-\gamma}, \quad \forall k \geq 1.$$

Now, for any $\epsilon > 0$, choosing $k$ such that $\frac{2\gamma^k}{(1-\gamma)^2} \leq \epsilon$, then, for any $m, n \geq k$ (assuming without loss of generality that $m \geq n$), we have

$$\left|V_{m,\mathcal{A}^-,\mathbf{0}}^{\text{Bayes},*}(s) - V_{n,\mathcal{A}^-,\mathbf{0}}^{\text{Bayes},*}\right| \leq \sum_{i=0}^{m-n-1}\left|V_{n+i+1,\mathcal{A}^-,\mathbf{0}}^{\text{Bayes},*} - V_{n+i,\mathcal{A}^-,\mathbf{0}}^{\text{Bayes},*}(s)\right|$$

$$\leq \sum_{i=0}^{m-n-1}\frac{2\gamma^{k+i}}{1-\gamma}$$

$$\leq \frac{2\gamma^k}{(1-\gamma)^2}$$

$$\leq \epsilon.$$

Therefore, for any $s \in \mathcal{S}$, $\{V_{+k}^*(s)\}_{k \geq 0}$ is a Cauchy sequence.

## E.2 Proof of Lemma E.2

To bound the value improvement from doubling the prediction window, we use a sub-Gaussian moment-generating function bound combined with a tailored Jensen-based analysis. For simplicity of presentation, we define an auxiliary $Q$-function, denoted by $\tilde{Q}_{+K}^*(s,\boldsymbol{a},\boldsymbol{\sigma})$. Specifically, for all $s \in \mathcal{S}$, $\boldsymbol{a} \in \mathcal{A}^K$, and $\boldsymbol{\sigma} \in \mathcal{Q}_K$,

$$\tilde{Q}_K^*(s,\boldsymbol{a},\boldsymbol{\sigma}) = \sum_{t=0}^{K-1}\gamma^t\left(\sum_{s_t}P(s_t|s,\boldsymbol{a}_{0:t-1},\boldsymbol{\sigma}_{1:t})r(s_t,a_t)\right) + \gamma^K\sum_{s_K}P(s_K|s,\boldsymbol{a},\boldsymbol{\sigma})V_{K,\mathcal{A}^-,\mathbf{0}}^{\text{Bayes},*}(s_K).$$

Thus, for any $s \in \mathcal{S}$ and $K \geq 1$, we have

$$V_{2K,\mathcal{A}^-,\mathbf{0}}^{\text{Bayes},*}(s) - V_{K,\mathcal{A}^-,\mathbf{0}}^{\text{Bayes},*}(s)$$

$$= \mathbb{E}_{\boldsymbol{\sigma}_{1:2K}}[\max_{\boldsymbol{a}} \tilde{Q}_{2K}^*(s, \boldsymbol{a}_{0:2K-1}, \boldsymbol{\sigma}_{1:2K})] - \mathbb{E}_{\boldsymbol{\sigma}_{1:K}}[\max_{\boldsymbol{a}} \tilde{Q}_K^*(s, \boldsymbol{a}_{0:K-1}, \boldsymbol{\sigma}_{1:K})]$$

$$= \int_{\mathcal{Q}_{2K}} \max_{\boldsymbol{a}} \tilde{Q}_{2K}^*(s, \boldsymbol{a}_{0:2K-1}, \boldsymbol{\sigma}_{1:2K}) P(d\boldsymbol{\sigma}_{1:2K})$$

$$\quad - \int_{\mathcal{Q}_K} \max_{\boldsymbol{a}} \tilde{Q}_K^*(s, \boldsymbol{a}_{0:K-1}, \boldsymbol{\sigma}_{1:K}) P(d\boldsymbol{\sigma}_{1:K})$$

$$= \int_{\boldsymbol{\sigma}_{1:K} \in \mathcal{Q}_K} \int_{\boldsymbol{\sigma}_{K+1:2K} \in \mathcal{Q}_K} \max_{\boldsymbol{a}} \tilde{Q}_{2K}^*(s, \boldsymbol{a}_{0:2K-1}, \boldsymbol{\sigma}_{1:2K}) P(d\boldsymbol{\sigma}_{1:K}) P(d\boldsymbol{\sigma}_{K1:2K})$$

$$\quad - \int_{\mathcal{Q}_K} \max_{\boldsymbol{a}} \tilde{Q}_K^*(s, \boldsymbol{a}_{0:K-1}, \boldsymbol{\sigma}_{1:K}) P(d\boldsymbol{\sigma}_{1:K})$$

$$= \int_{\boldsymbol{\sigma}_{1:K} \in \mathcal{Q}_K} \left( \int_{\boldsymbol{\sigma}_{K+1:2K} \in \mathcal{Q}_K} \left( \max_{\boldsymbol{a}} \tilde{Q}_{2K}^*(s, \boldsymbol{a}_{0:2K-1}, \boldsymbol{\sigma}_{1:2K}) \right. \right.$$

$$\left. \left. - \max_{\boldsymbol{a}} \tilde{Q}_K^*(s, \boldsymbol{a}_{0:K-1}, \boldsymbol{\sigma}_{1:K}) \right) P(d\boldsymbol{\sigma}_{K+1:2K}) \right) P(d\boldsymbol{\sigma}_{1:K}). \tag{25}$$

This treatment allows us to focus on the value improvement from additional transition prediction of $\boldsymbol{\sigma}_{K+1:2K}$. For any $\boldsymbol{\sigma}_{1:2K} \in \mathcal{Q}_{2K}$, we bound the term $\max_{\boldsymbol{a}} \tilde{Q}_{2K}^*(s, \boldsymbol{a}_{0:2K-1}, \boldsymbol{\sigma}_{1:2K}) - \max_{\boldsymbol{a}} \tilde{Q}_K^*(s, \boldsymbol{a}_{0:K-1}, \boldsymbol{\sigma}_{1:K})$ as follows:

$$\max_{\boldsymbol{a}} \tilde{Q}_{2K}^*(s, \boldsymbol{a}_{0:2K-1}, \boldsymbol{\sigma}_{1:2K}) - \max_{\boldsymbol{a}} \tilde{Q}_K^*(s, \boldsymbol{a}_{0:K-1}, \boldsymbol{\sigma}_{1:K})$$

$$= \max_{\boldsymbol{a}_{0:2K-1}} \left( \sum_{t=0}^{2K-1} \gamma^t \left( \sum_{s_t} P(s_t|s, \boldsymbol{a}_{0:t-1}, \boldsymbol{\sigma}_{1:t}) r(s_t, a_t) \right) + \gamma^{2K} \sum_{s_{2K}} P(s_{2K}|s, \boldsymbol{a}_{0:2K-1}, \boldsymbol{\sigma}_{1:2K}) V_{2K,\mathcal{A}^-,\mathbf{0}}^{\text{Bayes},*}(s_{2K}) \right)$$

$$\quad - \max_{\boldsymbol{a}_{0:K-1}} \left( \sum_{t=0}^{K-1} \gamma^t \left( \sum_{s_t} P(s_t|s, \boldsymbol{a}_{0:t-1}, \boldsymbol{\sigma}_{1:t}) r(s_t, a_t) \right) + \gamma^K \sum_{s_K} P(s_K|s, \boldsymbol{a}_{0:K-1}, \boldsymbol{\sigma}_{1:K}) V_{K,\mathcal{A}^-,\mathbf{0}}^{\text{Bayes},*}(s_K) \right)$$

$$= \max_{\boldsymbol{a}_{0:2K-1}} \left( \sum_{t=0}^{2K-1} \gamma^t \left( \sum_{s_t} P(s_t|s, \boldsymbol{a}_{0:t-1}, \boldsymbol{\sigma}_{1:t}) r(s_t, a_t) \right) + \gamma^{2K} \sum_{s_{2K}} P(s_{2K}|s, \boldsymbol{a}_{0:2K-1}, \boldsymbol{\sigma}_{1:2K}) V_{2K,\mathcal{A}^-,\mathbf{0}}^{\text{Bayes},*}(s_{2K}) \right)$$

$$\quad - \max_{\boldsymbol{a}_{0:2K-1}} \left( \sum_{t=0}^{2K-1} \gamma^t \left( \sum_{s_t} P(s_t|s, \boldsymbol{a}_{0:t-1}, \boldsymbol{\sigma}_{1:t}) r(s_t, a_t) \right) + \gamma^{2K} \sum_{s_{2K}} P(s_{2K}|s, \boldsymbol{a}_{0:2K-1}, \boldsymbol{\sigma}_{1:2K}) V_{K,\mathcal{A}^-,\mathbf{0}}^{\text{Bayes},*}(s_{2K}) \right)$$

$$\quad + \max_{\boldsymbol{a}_{0:2K-1}} \left( \sum_{t=0}^{2K-1} \gamma^t \left( \sum_{s_t} P(s_t|s, \boldsymbol{a}_{0:t-1}, \boldsymbol{\sigma}_{1:t}) r(s_t, a_t) \right) + \gamma^{2K} \sum_{s_{2K}} P(s_{2K}|s, \boldsymbol{a}_{0:2K-1}, \boldsymbol{\sigma}_{1:2K}) V_{K,\mathcal{A}^-,\mathbf{0}}^{\text{Bayes},*}(s_{2K}) \right)$$

$$\quad - \max_{\boldsymbol{a}_{0:K-1}} \left( \sum_{t=0}^{K-1} \gamma^t \left( \sum_{s_t} P(s_t|s, \boldsymbol{a}_{0:t-1}, \boldsymbol{\sigma}_{1:t}) r(s_t, a_t) \right) + \gamma^K \sum_{s_K} P(s_K|s, \boldsymbol{a}_{0:K-1}, \boldsymbol{\sigma}_{1:K}) V_{K,\mathcal{A}^-,\mathbf{0}}^{\text{Bayes},*}(s_K) \right). \tag{26}$$

Applying the max-difference inequality on Eq. (26), we can conclude:

$$\max_{\boldsymbol{a}} \tilde{Q}_{2K}^*(s, \boldsymbol{a}_{0:2K-1}, \boldsymbol{\sigma}_{1:2K}) - \max_{\boldsymbol{a}} \tilde{Q}_K^*(s, \boldsymbol{a}_{0:K-1}, \boldsymbol{\sigma}_{1:K})$$

$$\leq \gamma^{2K} \max_s (V_{2K,\mathcal{A}^-,\mathbf{0}}^{\text{Bayes},*}(s) - V_{K,\mathcal{A}^-,\mathbf{0}}^{\text{Bayes},*}(s))$$

$$\quad + \gamma^K \max_{\boldsymbol{a}_{0:2K-1}} \left( \sum_{t=K}^{2K-1} \gamma^{t-K} \left( \sum_{s_t} P(s_t|\boldsymbol{a}_{K:t}, \boldsymbol{\sigma}_{K+1:t+1}, s_K) r(s_t, a_t) \right) \right)$$

$$+\gamma^K \sum_{s_{2K}} P(s_{2K}|\boldsymbol{a}_{0:2K-1}, \boldsymbol{\sigma}_{1:2K}) V_{K,\mathcal{A}^-,\boldsymbol{0}}^{\text{Bayes},*}(s_{2K}) - \sum_{s_K} P(s_K|s, \boldsymbol{a}_{0:K-1}, \boldsymbol{\sigma}_{1:K}) V_{K,\mathcal{A}^-,\boldsymbol{0}}^{\text{Bayes},*}(s_K) \Bigg).$$
(27)

Combining Eq. (25) and (27), and take the maximal on states $s$ on both sides, we have a simplified bound in a trajectory-wise reward form:

$$\max_s \left( V_{2K,\mathcal{A}^-,\boldsymbol{0}}^{\text{Bayes},*}(s) - V_{K,\mathcal{A}^-,\boldsymbol{0}}^{\text{Bayes},*}(s) \right) \leq \frac{\gamma^K}{1-\gamma^{2K}} \max_s \mathbb{E}_{\boldsymbol{\sigma}_{K+1:2K}} \left[ \max_{\boldsymbol{a}_{K:2K-1}} (\boldsymbol{P}_s(\boldsymbol{\sigma}_{K+1:2K}) - \overline{\boldsymbol{P}}_s) \boldsymbol{V}_s^* \right],$$
(28)

where $\boldsymbol{V}_s^* \in \mathbb{R}^{(|\mathcal{S}||\mathcal{A}|)^K}$ is a trajectory-wise reward vector, defined as:

$$\boldsymbol{V}_s^*(s_{K+1}, s_{K+2}, ..., s_{2K}, a_K, a_{K+1}, ..., a_{2K-1}) = \sum_{t=K}^{2K-1} \gamma^{t-K} r(s_t, a_t) + \gamma^K V_{K,\mathcal{A}^-,\boldsymbol{0}}^{\text{Bayes},*}(s_{2K}).$$

The matrix $\boldsymbol{P}_s(\boldsymbol{\sigma}_{K+1:2K}) \in \mathbb{R}^{|\mathcal{A}|^K \times (|\mathcal{S}||\mathcal{A}|)^K}$, with each entry denote the probability of visiting a state-action trajectory $(s_{K+1}, s_{K+2}, ..., s_{2K}, a_K, a_K, ..., a_{2K-1})$ given initial state $s$, the action vector $\boldsymbol{a}_{K:2K-1}$ and transition $\boldsymbol{\sigma}_{K+1:2K}$. The matrix $\boldsymbol{P}_s(\boldsymbol{\sigma}_{K+1:2K}) = \mathbb{E}_{\boldsymbol{\sigma}_{K+1:2K}}[\boldsymbol{P}_s(\boldsymbol{\sigma}_{K+1:2K})]$.

Let $X_{s,\boldsymbol{a}_{K+1:2K},\boldsymbol{\sigma}_{K+1:2K}} = (\boldsymbol{P}_s(\boldsymbol{\sigma}_{K+1:2K}) - \overline{\boldsymbol{P}}_s)\boldsymbol{V}_s^*|_{\boldsymbol{a}_{K:2K-1}}$, which denotes the $\boldsymbol{a}_{K:2K-1}$-th entry of the vector $(\boldsymbol{P}_s(\boldsymbol{\sigma}_{K+1:2K}) - \overline{\boldsymbol{P}}_s)\boldsymbol{V}_s^*$. We can verify that $X_{s,\boldsymbol{a}_{K+1:2K},\boldsymbol{\sigma}_{K+1:2K}}$ is the cumulative discounted reward, whose absolute value is bounded by $\frac{1}{1-\gamma}$. For simplicity, we denote $X_{s,\boldsymbol{a}_{K+1:2K},\boldsymbol{\sigma}_{K+1:2K}}$ by $X_{s,\boldsymbol{a},\boldsymbol{\sigma}}$. Therefore, $X_{s,\boldsymbol{a},\boldsymbol{\sigma}}$ is a sub-Gaussian random variable, which implies

$$\mathbb{E}_{\boldsymbol{\sigma}}(e^{\lambda X_{s,\boldsymbol{a},\boldsymbol{\sigma}}}) \leq e^{\frac{\theta_{s,\boldsymbol{a},K}^2 \lambda^2}{2}}, \quad \forall \boldsymbol{a} \in \mathcal{A}^K,$$

Where $\theta_{s,\boldsymbol{a},K}^2$ denotes the sub-Gaussian parameter [58] of $X_{s,\boldsymbol{a},\boldsymbol{\sigma}}$. Therefore, using Jensen's inequality and the monotonicity of exponential functions, we have for all $\lambda$:

$$e^{\lambda \mathbb{E}_{\boldsymbol{\sigma}}(\max_{\boldsymbol{a} \in \mathcal{A}^K} X_{s,\boldsymbol{a},\boldsymbol{\sigma}})} \leq \mathbb{E}_{\boldsymbol{\sigma}}(\max_{\boldsymbol{a} \in \mathcal{A}^K} e^{\lambda X_{s,\boldsymbol{a},\boldsymbol{\sigma}}}) \leq \sum_{\boldsymbol{a} \in \mathcal{A}^K} \mathbb{E}(e^{\lambda X_{s,\boldsymbol{a},\boldsymbol{\sigma}}}) \leq \sum_{\boldsymbol{a} \in \mathcal{A}^K} e^{\frac{\theta_{s,\boldsymbol{a},K}^2 \lambda^2}{2}}.$$

Taking the logarithm on both sides of the previous inequality, we have

$$
\begin{aligned}
\mathbb{E}_{\boldsymbol{\sigma}} \left( \max_{\boldsymbol{a} \in \mathcal{A}^K} X_{s,\boldsymbol{a},\boldsymbol{\sigma}} \right) &\leq \frac{\log \left( \sum_{\boldsymbol{a} \in \mathcal{A}^K} e^{\frac{\theta_{s,\boldsymbol{a},K}^2 \lambda^2}{2}} \right)}{\lambda} \\
&\leq \frac{\log \left( |\mathcal{A}|^K e^{\frac{\max_{\boldsymbol{a}} \theta_{s,\boldsymbol{a},K}^2 \lambda^2}{2}} \right)}{\lambda} \\
&= \frac{K \log |\mathcal{A}| + \frac{\max_{\boldsymbol{a}} \theta_{s,\boldsymbol{a},K}^2 \lambda^2}{2}}{\lambda}.
\end{aligned}
$$
(29)

By choosing $\lambda = \sqrt{\frac{2K \log |\mathcal{A}|}{\max_{\boldsymbol{a}} \theta_{s,\boldsymbol{a},K}^2}}$ in Eq. (29), we have

$$
\begin{aligned}
\mathbb{E}_{\boldsymbol{\sigma}_{K+1:2K}} \left[ \max_{\boldsymbol{a}_{K:2K-1}} (\boldsymbol{P}_s(\boldsymbol{\sigma}_{K+1:2K}) - \overline{\boldsymbol{P}}_s)\boldsymbol{V}_s^* \right] &\leq \frac{K \log |\mathcal{A}| + \frac{\max_{\boldsymbol{a}} \theta_{s,\boldsymbol{a},K}^2 \cdot \frac{2K \log |\mathcal{A}|}{\max_{\boldsymbol{a}} \theta_{s,\boldsymbol{a},K}^2}}{2}}{\sqrt{\frac{2K \log |\mathcal{A}|}{\max_{\boldsymbol{a}} \theta_{s,\boldsymbol{a},K}^2}}} \\
&= \sqrt{2K \max_{\boldsymbol{a}} \theta_{s,\boldsymbol{a},K}^2 \log |\mathcal{A}|}.
\end{aligned}
$$
(30)

Substituting Eq. (30) into Eq. (28) yields:

$$\max_s \left( V_{2K,\mathcal{A}^-,\boldsymbol{0}}^{\text{Bayes},*}(s) - V_{K,\mathcal{A}^-,\boldsymbol{0}}^{\text{Bayes},*}(s) \right) \leq \frac{\gamma^K \sqrt{2K \max_{s,\boldsymbol{a}} \theta_{s,\boldsymbol{a},K}^2 \log |\mathcal{A}|}}{1-\gamma^{2K}}, \quad \forall s \in \mathcal{S}, K \geq 1.$$
(31)

Due to the boundedness of the reward function, the sub-Gaussian $\max_{s,a} \theta^2_{s,a,K}$ of trajectory can be upper bounded by:

$$\max_{s,a} \theta^2_{s,a,K} \le C(1-\gamma)^{-2}, \tag{32}$$

where $C$ is an absolute constant.

Combining (32) and (31) yields our result.

### E.3 Proof of Lemma E.3

Using Lemma E.2 in Eq. (20) and , we have:

$$
\begin{aligned}
V^{\mathrm{Bayes},*}_{\infty,\mathcal{A}^-,\mathbf{0}}(s) - V^{\mathrm{Bayes},*}_{K,\mathcal{A}^-,\mathbf{0}}(s) &= \lim_{k\to\infty}\left(\sum_{i=0}^{k-1}(V^{\mathrm{Bayes},*}_{K\cdot 2^{i+1},\mathcal{A}^-,\mathbf{0}} - V^{\mathrm{Bayes},*}_{K\cdot 2^i,\mathcal{A}^-,\mathbf{0}})\right) \\
&\le \lim_{k\to\infty}\left(\sum_{i=0}^{k-1}\frac{\gamma^{K\cdot 2^i}\sqrt{CK\cdot 2^i \log|\mathcal{A}|}}{(1-\gamma)(1-\gamma^{2K\cdot 2^i})}\right) \\
&= \sum_{i=0}^{\infty}\frac{\gamma^{K\cdot 2^i}\sqrt{CK\cdot 2^i \log|\mathcal{A}|}}{(1-\gamma)(1-\gamma^{2K\cdot 2^i})} \\
&\le \frac{\sqrt{CK\log|\mathcal{A}|}}{(1-\gamma)(1-\gamma^{2K})}\sum_{i=0}^{\infty}\left(\gamma^{K\cdot 2^i}\sqrt{2^i}\right). \tag{33}
\end{aligned}
$$

Next, we focus on bounding the term $\sum_{i=0}^{\infty}(\gamma^{K\cdot 2^i}\sqrt{2^i})$. Denote $a_i = \gamma^{K\cdot 2^i}\sqrt{2^i}$. Consider the index $i^* = \frac{7}{20}\log_2(\frac{1}{1-\gamma})$, for any $i \ge i^*$, we notice that:

$$\gamma^{K\cdot 2^i} = (\gamma^{2^i})^K \le \left(\gamma^{\frac{1}{1-\gamma}}\right)^{\frac{7}{20}\cdot K} \le \left(\gamma^{\frac{1}{1-\gamma}}\right)^{\frac{7}{20}} \le e^{-7/20}, \quad \forall \gamma \in [0,1).$$

Hence, for any $i \ge i^*$, we have:

$$a_{i+1} = \gamma^{K2^{i+1}}\sqrt{2^{i+1}} = \gamma^{K\cdot 2^i}\cdot\gamma^{K\cdot 2^i}\cdot\sqrt{2^{i+1}} \le e^{-7/20}\cdot\sqrt{2}\gamma^{K\cdot 2^i}\sqrt{2^i} \le \frac{997}{1000}a_i.$$

Summing $a_i$ from $i = \lceil i^* \rceil$ to infinity yields that:

$$\sum_{i=\lceil i^*\rceil}^{\infty} a_i \le \sum_{i=\lceil i^*\rceil}^{\infty}\left(\frac{997}{1000}\right)^{i-\lceil i^*\rceil} a_i \le \frac{1000}{3}a_{i^*} \le \frac{1000}{3}\gamma^K\sqrt{2^{i^*}} \le \frac{1000\gamma^K}{3(1-\gamma)^{\frac{7}{40}}}.$$

For the sum of $a_i$ from $i = 0$ to $\lceil i^* \rceil - 1$, we have:

$$\sum_{i=0}^{\lceil i^*\rceil-1} a_i = \sum_{i=0}^{\lceil i^*\rceil-1}\gamma^{K\cdot 2^i}\sqrt{2^i} \le \gamma^K\sum_{i=0}^{\lceil i^*\rceil-1}\sqrt{2^i} \le 8\gamma^K\sqrt{2^{i^*}} \le \frac{8\gamma^K}{(1-\gamma)^{\frac{7}{40}}}.$$

Then the desired sum satisfies:

$$\sum_{i=0}^{\infty} a_i \le \sum_{i=0}^{\lceil i^*\rceil-1} a_i + \sum_{i=\lceil i^*\rceil}^{\infty} a_i \le \frac{1024\gamma^K}{3(1-\gamma)^{\frac{7}{40}}}.$$

Hence, we can conclude that:

$$\max_s\left(V^*_{+\infty}(s) - V^*_{+K}(s)\right) \le \frac{C_1\gamma^K\sqrt{K\log|\mathcal{A}|}}{(1-\gamma)^{\frac{47}{40}}(1-\gamma^{2K})} \le \frac{C_1\gamma^K\sqrt{K\log|\mathcal{A}|}}{(1-\gamma)^{\frac{6}{5}}(1-\gamma^{2K})},$$

where $C_1$ is an absolute constant.

## E.4 Proof of Lemma E.4

The desired gap satisfies:

$$
V_{\text{off}}^{\text{Bayes},*}(s) - V_{\infty,\mathcal{A}^-,\mathbf{0}}^{\text{Bayes},*}(s) = \lim_{k\to\infty} \mathbb{E}_{\boldsymbol{\sigma}_{1:k}^*} \left( \max_{\boldsymbol{a}_{0:k-1}} \left( \sum_{t=0}^{k-1} \gamma^t r(s_t, a_t | s_0 = s, \boldsymbol{\sigma}_{1:t}^*) \right) \right.
$$

$$
\left. - \max_{\boldsymbol{a}_{0:k-1}} \left( \sum_{t=0}^{k-1} \gamma^t r(s_t, a_t | s_0 = s, \boldsymbol{\sigma}_{1:t}) \right) \right)
$$

$$
\leq \lim_{k\to\infty} \mathbb{E}_{\boldsymbol{\sigma}_{1:k}^*} \left( \max_{\boldsymbol{a}_{0:k-1}} \left( \sum_{t=0}^{k-1} \gamma^t (r(s_t, a_t | \boldsymbol{\sigma}_{1:t}^*) - r(s_t, a_t | \boldsymbol{\sigma}_{1:t})) \right) \right)
$$

$$
\leq \lim_{k\to\infty} \sum_{t=0}^{k-1} \underbrace{\mathbb{E}_{\boldsymbol{\sigma}_{1:t}^*} \left( \max_{\boldsymbol{a}_{0:t}} \gamma^t (r(s_t, a_t | \boldsymbol{\sigma}_{1:t}^*) - r(s_t, a_t | \boldsymbol{\sigma}_{1:t})) \right)}_{Q_t},
$$

where the last inequality follows by exchanging the limit and expectation with the finite summation. Then we only need to bound each $Q_t$ separately. Specifically, for each $Q_t$, it can be rewritten into:

$$
Q_t = \mathbb{E}_{\boldsymbol{\sigma}_{1:t}^*} \left( \max_{\boldsymbol{a}_{0:t}} \gamma^t (r(s_t, a_t | \boldsymbol{\sigma}_{1:t}^*) - r(s_t, a_t | \boldsymbol{\sigma}_{1:t})) \right)
$$

$$
= \gamma^t \mathbb{E}_{\boldsymbol{\sigma}_{1:t}^*} \left( \max_{\boldsymbol{a}_{0:t}} (r(s_t, a_t | \boldsymbol{\sigma}_{1:t}^*) - r(s_t, a_t | \boldsymbol{\sigma}_{1:t})) \right)
$$

$$
= \gamma^t \mathbb{E}_{\boldsymbol{\sigma}_{1:t}^*} \left( \max_{\boldsymbol{a}_{0:t}} \left( \sum_{s_t} (P(s_t | \boldsymbol{\sigma}_{1:t}^*, \boldsymbol{a}_{0:t-1}) - P(s_t | \boldsymbol{\sigma}_{1:t}, \boldsymbol{a}_{0:t-1})) \, r(s_t, a_t) \right) \right)
$$

$$
= \gamma^t \mathbb{E}_{\boldsymbol{\sigma}_{1:t}^*} \left( \max_{\boldsymbol{a}_{0:t}} (\boldsymbol{P}_{s,\boldsymbol{\sigma}_{1:t}^*} - \boldsymbol{P}_{s,\boldsymbol{\sigma}_{1:t}}) \boldsymbol{r} \right), \tag{34}
$$

where $\boldsymbol{r} \in \mathbb{R}^{|\mathcal{S}||\mathcal{A}|}$ denotes the reward vector, with the $(s,a)$-th entry equal to $r(s,a)$; $\boldsymbol{P}_{s,\boldsymbol{\sigma}_{1:t}^*} \in \mathbb{R}^{|\mathcal{A}|^{t+1} \times |\mathcal{S}||\mathcal{A}|}$ and $\boldsymbol{P}_{s,\boldsymbol{\sigma}_{1:t}} \in \mathbb{R}^{|\mathcal{A}|^{t+1} \times |\mathcal{S}||\mathcal{A}|}$ denote the random transition probability matrices from the initial state $s$ to the state-action pair $(s_t, a_t)$ under the action sequence $\boldsymbol{a}_{0:t}$, specified by the transition predictions $\boldsymbol{\sigma}_{1:t}^*$ and $\boldsymbol{\sigma}_{1:t}$, respectively.

Note that $\boldsymbol{P}_{s,\boldsymbol{\sigma}_{1:t}}$ differs from $\boldsymbol{P}_{s,\boldsymbol{\sigma}_{1:t}^*}$ only due to partial predictability. We can observe that the form in Eq. (34) is with the similar form as Eq. (28) in Appendix E.2. We can follow the same routine in the proof to Lemma E.2 in Appendix E.2 to handle Eq. (34). We can verify that for any action sequence $\boldsymbol{a}_{0:t} \in (\mathcal{A}^-)^{t+1}$, the corresponding rows of $\boldsymbol{P}_{s,\boldsymbol{\sigma}_{1:t}}$ and $\boldsymbol{P}_{s,\boldsymbol{\sigma}_{1:t}^*}$ are identical. Formally, for all $\boldsymbol{\sigma}_{1:t}^*$, we have

$$
\left[ (\boldsymbol{P}_{s,\boldsymbol{\sigma}_{1:t}^*} - \boldsymbol{P}_{s,\boldsymbol{\sigma}_{1:t}}) \boldsymbol{r} \right]_{\boldsymbol{a}_{0:t}} = 0, \quad \forall \boldsymbol{a}_{0:t} \in (\mathcal{A}^-)^{t+1}. \tag{35}
$$

Hence, under any fixed $t$, the two $t$-step kernels $P_{\text{off},1:t}$ and $P_{+,1:t}$ agree on every row corresponding to an action sequence in $(\mathcal{A}^-)^{t+1}$. Since there are $|\mathcal{A}^-|^{t+1}$ such sequences, the number of remaining mismatched rows is $|\mathcal{A}|^{t+1} - |\mathcal{A}^-|^{t+1}$. Adding the trivial all-zero case yields at most $|\mathcal{A}|^{t+1} - |\mathcal{A}^-|^{t+1} + 1$ distinct nonzero differences between the two kernels.

Applying Lemma E.2 to each such mismatched row, we obtain for all $t \geq 1$:

$$
Q_t \leq \gamma^t C_2 \sqrt{\theta_{\max}^2 \ln(|\mathcal{A}|^{t+1} - |\mathcal{A}^-|^{t+1} + 1)}, \tag{36}
$$

where $C_2$ is an absolute constant and $\theta_{\max}^2 = \max_{s,\boldsymbol{a}_{0:t},t} \sigma^2(r(s_t, a_t | s_0 = s, \boldsymbol{a}_{0:t}))$, where $\sigma(\cdot)$ denotes the sub-Gaussian parameter. Parameter $\theta_{\max}^2$ indicates the maximal variance of the reward.

Summing over $t = 1, 2, \ldots$ for Eq. (36) then gives

$$
\max_{s\in\mathcal{S}} (V_{\text{off}}^{\text{Bayes},*}(s) - V_{\infty,\mathcal{A}^-,\mathbf{0}}^{\text{Bayes},*}(s)) \leq C_2 \sum_{t=1}^{\infty} \gamma^t \sqrt{\theta_{\max}^2 \ln(|\mathcal{A}|^{t+1} - |\mathcal{A}^-|^{t+1} + 1)}.
$$

This completes the proof.

## E.5 Proof of Lemma E.5

For clarity of presentation, for any Bayesian value function $V^{\text{Bayes}} \in \mathbb{R}^{|\mathcal{S}|}$ and $K$-step transition prediction $\boldsymbol{\sigma} \in \mathcal{Q}_K$, we define:

$$R(s, V^{\text{Bayes}}, \boldsymbol{\sigma}) = \max_{\boldsymbol{a}} \left( \sum_{t=0}^{K-1} \gamma^t \left( \sum_{s_t} P(s_t|s, \boldsymbol{a}_{0:t-1}, \boldsymbol{\sigma}_{1:t}) r(s_t, a_t) \right) \right.$$
$$\left. + \gamma^K \sum_{s_K} P(s_K|s, \boldsymbol{a}_{0:K-1}, \boldsymbol{\sigma}) V^{\text{Bayes}}(s_K) \right). \tag{37}$$

Applying the Bayesian Bellman equation on $T_3$, we have:

$$V_{K,\mathcal{A}^-,\boldsymbol{0}}^{\text{Bayes},*}(s) - V_{K,\mathcal{A}^-,\boldsymbol{\varepsilon}}^{\text{Bayes},*}(s) = \mathbb{E}_{\boldsymbol{\sigma} \sim \mathcal{P}_{1:K}^*} \left[ R(s, V_{K,\mathcal{A}^-,\boldsymbol{0}}^{\text{Bayes},*}, \boldsymbol{\sigma}) \right] - \mathbb{E}_{\boldsymbol{\sigma} \sim \widehat{\mathcal{P}}_{1:K}} \left[ R(s, V_{K,\mathcal{A}^-,\boldsymbol{\varepsilon}}^{\text{Bayes},*}, \boldsymbol{\sigma}) \right]$$
$$= \mathbb{E}_{\boldsymbol{\sigma} \sim \mathcal{P}_{1:K}^*} \left[ R(s, V_{K,\mathcal{A}^-,\boldsymbol{0}}^{\text{Bayes},*}, \boldsymbol{\sigma}) \right] - \mathbb{E}_{\boldsymbol{\sigma} \sim \widehat{\mathcal{P}}_{1:K}} \left[ R(s, V_{K,\mathcal{A}^-,\boldsymbol{0}}^{\text{Bayes},*}, \boldsymbol{\sigma}) \right]$$
$$+ \mathbb{E}_{\boldsymbol{\sigma} \sim \widehat{\mathcal{P}}_{1:K}} \left[ R(s, V_{K,\mathcal{A}^-,\boldsymbol{0}}^{\text{Bayes},*}, \boldsymbol{\sigma}) \right] - \mathbb{E}_{\boldsymbol{\sigma} \sim \widehat{\mathcal{P}}_{1:K}} \left[ R(s, V_{K,\mathcal{A}^-,\boldsymbol{\varepsilon}}^{\text{Bayes},*}, \boldsymbol{\sigma}) \right],$$

where $\mathcal{P}_{1:K}^*$ and $\widehat{\mathcal{P}}_{1:K}$ denote the distributions of accurate and inaccurate $K$-step transition prediction $\boldsymbol{\sigma}$.

Taking the absolute value on both sides and selecting the state $s$ that maximizes it yields:

$$\max_s |V_{K,\mathcal{A}^-,\boldsymbol{0}}^{\text{Bayes},*}(s) - V_{K,\mathcal{A}^-,\boldsymbol{\varepsilon}}^{\text{Bayes},*}(s)| \leq \max_s \left| \mathbb{E}_{\boldsymbol{\sigma} \sim \mathcal{P}_{1:K}^*} \left[ R(s, V_{K,\mathcal{A}^-,\boldsymbol{0}}^{\text{Bayes},*}, \boldsymbol{\sigma}) \right] - \mathbb{E}_{\boldsymbol{\sigma} \sim \widehat{\mathcal{P}}_{1:K}} \left[ R(s, V_{K,\mathcal{A}^-,\boldsymbol{0}}^{\text{Bayes},*}, \boldsymbol{\sigma}) \right] \right|,$$
$$+ \gamma^K \max_s |V_{K,\mathcal{A}^-,\boldsymbol{0}}^{\text{Bayes},*}(s) - V_{K,\mathcal{A}^-,\boldsymbol{\varepsilon}}^{\text{Bayes},*}(s)|$$
$$\leq \frac{\max_s \left| \mathbb{E}_{\boldsymbol{\sigma} \sim \mathcal{P}_{1:K}^*} \left[ R(s, V_{K,\mathcal{A}^-,\boldsymbol{0}}^{\text{Bayes},*}, \boldsymbol{\sigma}) \right] - \mathbb{E}_{\boldsymbol{\sigma} \sim \widehat{\mathcal{P}}_{1:K}} \left[ R(s, V_{K,\mathcal{A}^-,\boldsymbol{0}}^{\text{Bayes},*}, \boldsymbol{\sigma}) \right] \right|}{1 - \gamma^K}.$$

Now we focus on $\left| \mathbb{E}_{\boldsymbol{\sigma} \sim \mathcal{P}_{1:K}^*} \left[ R(s, V_{K,\mathcal{A}^-,\boldsymbol{0}}^{\text{Bayes},*}, \boldsymbol{\sigma}) \right] - \mathbb{E}_{\boldsymbol{\sigma} \sim \widehat{\mathcal{P}}_{1:K}} \left[ R(s, V_{K,\mathcal{A}^-,\boldsymbol{0}}^{\text{Bayes},*}, \boldsymbol{\sigma}) \right] \right|$. The difference in this term comes from the prediction errors, which we use the Kantorovich-Rubinstein inequality to bound:

**Lemma E.6** (Kantorovich-Rubinstein Inequality [59]). *Let $(\mathcal{X}, d)$ be a Polish metric space, and let $\mu, \nu$ be two probability measures over $\mathcal{X}$. Let $f : \mathcal{X} \to \mathbb{R}$ be a measurable function that is $L$-Lipschitz with respect to $d$, i.e.,*

$$|f(x) - f(y)| \leq L \cdot d(x, y), \quad \forall x, y \in \mathcal{X}.$$

*Then the difference in expectations satisfies*

$$|\mathbb{E}_\mu[f] - \mathbb{E}_\nu[f]| \leq L \cdot W_1(\mu, \nu),$$

*where $W_1(\mu, \nu)$ is the Wasserstein-1 distance defined by*

$$W_1(\mu, \nu) := \inf_{\gamma \in \Pi(\mu, \nu)} \int_{\mathcal{X} \times \mathcal{X}} d(x, y) \, d\gamma(x, y),$$

*and $\Pi(\mu, \nu)$ denotes the set of all couplings (joint distributions) with marginals $\mu$ and $\nu$.*

We first show the perturbation sensitivity. A perturbation in $\sigma_j$ changes only the terms involving $\{s_t\}$ for $t \geq j$. Its impact on the reward-sum $\sum_{t=j}^{K-1} \gamma^t r(s_t, a_t)$ is bounded by $\sum_{t=j}^{K-1} \gamma^t \leq \frac{\gamma^j - \gamma^K}{1-\gamma}$, while its impact on the terminal term $\gamma^K V_{K,\mathcal{A}^-,\boldsymbol{0}}^{\text{Bayes},*}(s_K)$ is $\gamma^K \max_s |V_{K,\mathcal{A}^-,\boldsymbol{0}}^{\text{Bayes},*}(s)| \leq \frac{\gamma^K}{1-\gamma}$. Hence, for each $j$,

$$\left| R(s, V_{K,\mathcal{A}^-,\boldsymbol{0}}^{\text{Bayes},*}, \boldsymbol{\sigma}) - R(s, V_{K,\mathcal{A}^-,\boldsymbol{0}}^{\text{Bayes},*}, \boldsymbol{\sigma}') \right| \leq \sum_{j=1}^K \left( \frac{\gamma^j - \gamma^K}{1-\gamma} + \frac{\gamma^K}{1-\gamma} \right) W_1(\sigma_j, \sigma_j') = \sum_{j=1}^K \frac{\gamma^j W_1(\sigma_j, \sigma_j')}{1-\gamma}.$$

Hence, we introduce the distance metric $d(\boldsymbol{\sigma}, \boldsymbol{\sigma}') = \sum_{j=1}^{K} \frac{\gamma^j W_1(\sigma_j, \sigma'_j)}{1-\gamma}$. Under $d$, the above inequality simply says $\left| R(s, V_{K,\mathcal{A}^-,\mathbf{0}}^{\text{Bayes},*}, \boldsymbol{\sigma}) - R(s, V_{K,\mathcal{A}^-,\mathbf{0}}^{\text{Bayes},*}, \boldsymbol{\sigma}') \right| \leq d(\boldsymbol{\sigma}, \boldsymbol{\sigma}')$, and $R$ is 1-Lipschitz.

By the KantorovichRubinstein inequality in Lemma E.6, for two measures $\mathcal{P}_{1:K}^*, \widehat{\mathcal{P}}_{1:K}$ on the product space,

$$\left| \mathbb{E}_{\boldsymbol{\sigma} \sim \mathcal{P}_{1:K}^*} \left[ R(s, V_{K,\mathcal{A}^-,\mathbf{0}}^{\text{Bayes},*}, \boldsymbol{\sigma}) \right] - \mathbb{E}_{\boldsymbol{\sigma} \sim \widehat{\mathcal{P}}_{1:K}} \left[ R(s, V_{K,\mathcal{A}^-,\mathbf{0}}^{\text{Bayes},*}, \boldsymbol{\sigma}) \right] \right| \leq W_1^{(d)}(\mathcal{P}_{1:K}^*, \widehat{\mathcal{P}}_{1:K}).$$

Since $\mathcal{P}_{1:K}^* = \bigotimes_{j=1}^K \mathcal{P}_j^*$ and $\widehat{\mathcal{P}}_{1:K} = \bigotimes_{j=1}^K \widehat{\mathcal{P}}_j$, where $\mathcal{P}_j^*$ and $\widehat{\mathcal{P}}_j$ denote the distribution of $\sigma_j$ when prediction is accurate and inaccurate, respectively. One checks:

$$W_1^{(d)}(\mathcal{P}_{1:K}^*, \widehat{\mathcal{P}}_{1:K}) \leq \sum_{j=1}^K \frac{\gamma^j}{1-\gamma} W_1^{(d)}(\mathcal{P}_j^*, \widehat{\mathcal{P}}_j) = \sum_{j=1}^K \frac{\gamma^j}{1-\gamma} \epsilon_j.$$

Thus, we have:

$$\max_s |V_{K,\mathcal{A}^-,\mathbf{0}}^{\text{Bayes},*}(s) - V_{K,\mathcal{A}^-,\boldsymbol{\varepsilon}}^{\text{Bayes},*}(s)| \leq \sum_{j=1}^K \frac{\gamma^j}{(1-\gamma)(1-\gamma^K)} \epsilon_j. \tag{38}$$

# F   Proof of Theorem 5.1

Based on the proposed algorithm, the required sample amounts $D_1$ and $D_2$ satisfy:

$$D_1 = N_1|\mathcal{S}|(|\mathcal{A}| - |\mathcal{A}^-|) + |\mathcal{S}||\mathcal{A}|, \quad D_2 = N_2. \tag{39}$$

The first term $D_1$ represents the sample complexity of estimating the environment model. The term $N_1|\mathcal{S}|(|\mathcal{A}| - |\mathcal{A}^-|)$ denotes the learning cost transition kernel $\widehat{P}(s'|s,a)$ for $s \in \mathcal{S}$ and $a \in \mathcal{A} \setminus \mathcal{A}^-$, where $|\mathcal{S}|(|\mathcal{A}| - |\mathcal{A}^-|)$ denotes the number of sampled entries, and $N_1$ denotes the number of times each entry is sampled. Learning the cost function only requires to sample each state action pair once with $|\mathcal{S}||\mathcal{A}|$ samples. The second term $D_1$ directly equals the number of samples $N_2$ from the prediction oracle.

We need to determine appropriate values for $N_1$ and $N_2$ to ensure that the estimation error of $\widehat{V}_{K,\mathcal{A}^-,\varepsilon}^{\text{Bayes},*}$ is smaller than $\epsilon$.

Let's first analyze the structure of the error. For any state $s \in \mathcal{S}$, the estimate Bayesian value function $\widehat{V}_{K,\mathcal{A}^-,\varepsilon}^{\text{Bayes},*}(s)$ satisfies:

$$\widehat{V}_{K,\mathcal{A}^-,\varepsilon}^{\text{Bayes},*}(s) = \int_{\boldsymbol{\sigma}_{1:K} \in \mathcal{Q}_K} \widehat{P}(d\boldsymbol{\sigma}_{1:K}) \max_{\boldsymbol{a}} \left( \sum_{t=0}^{K-1} \gamma^t \left( \sum_{s_t} \widehat{P}(s_t|s, \boldsymbol{a}_{0:t-1}, \boldsymbol{\sigma}_{1:t}) r(s_t, a_t) \right) \right.$$
$$\left. + \gamma^K \sum_{s_K} \widehat{P}(s_K|s, \boldsymbol{a}, \boldsymbol{\sigma}) \widehat{V}_{K,\mathcal{A}^-,\varepsilon}^{\text{Bayes},*}(s_K) \right), \tag{40}$$

where the estimated multi-step transition kernel $\widehat{P}(s_t|s, \boldsymbol{a}_{0:t-1}, \boldsymbol{\sigma}_{1:t})$ satisfies the following recursive condition:

$$\widehat{P}(s_t|s, \boldsymbol{a}_{0:t-1}, \boldsymbol{\sigma}_{1:t}) = \sum_{s_{t-1}} \widehat{P}(s_t|s_{t-1}, a_{t-1}, \sigma_t) \widehat{P}_{t-1}(s_{t-1}|s, \boldsymbol{a}_{0:t-2}, \boldsymbol{\sigma}_{1:t-1}), \forall t \leq K.$$

And $\widehat{P}(s_t|s_{t-1}, a_{t-1}, \sigma_t)$ satisfies:

$$\widehat{P}(s_t|s_{t-1}, a_{t-1}, \sigma_t) = \begin{cases} \sigma_t((s_{t-1}, a_{t-1}), s_t), a \in \mathcal{A}^-, \\ \widehat{P}(s_t|s_{t-1}, a_{t-1}), a \notin \mathcal{A}^-. \end{cases} \tag{41}$$

We can see that, any term $\widehat{P}(s_t|s, \boldsymbol{a}_{0:t-1}, \boldsymbol{\sigma}_{1:t})$ presents finite-sample error only caused by the estimation error of $\widehat{P}(s_{t'}|s_{t'-1}, a_{t'-1})$ with $t' \leq t$. And the Bayesian value function estimation

$\widehat{V}_{K,\mathcal{A}^-,\varepsilon}^{\text{Bayes},*}(s)$ is influenced by the estimation errors of both $\widehat{P}(s_t|s_{t-1},a_{t-1})$ and $\widehat{P}(\boldsymbol{\sigma})$. To highlight such dependence, we define an auxiliary value function $V(P_{\boldsymbol{\sigma}}, P_{(s,a)}, V^{\text{Bayes}}, s)$ as

$$
V(P_{\boldsymbol{\sigma}}, P_{(s,a)}, V^{\text{Bayes}}, s)
$$
$$
= \int_{\boldsymbol{\sigma}_{1:K} \in \mathcal{Q}_K} P_{\boldsymbol{\sigma}}(d\boldsymbol{\sigma}_{1:K}) \max_{\boldsymbol{a}} \left( \sum_{t=0}^{K-1} \gamma^t \left( \sum_{s_t} P_{(s,a)}(s_t|s, \boldsymbol{a}_{0:t-1}, \boldsymbol{\sigma}_{1:t}) r(s_t, a_t) \right) \right.
$$
$$
\left. + \gamma^K \sum_{s_K} P_{(s,a)}(s_K|s, \boldsymbol{a}, \boldsymbol{\sigma}) V^{\text{Bayes}}(s_K) \right), s_0 = s. \tag{42}
$$

Hence, $\widehat{V}_{K,\mathcal{A}^-,\varepsilon}^{\text{Bayes},*}(s)$ can be denoted by $\widehat{V}_{K,\mathcal{A}^-,\varepsilon}^{\text{Bayes},*}(s) = V(\widehat{P}(\boldsymbol{\sigma}), \widehat{P}(s'|s,a), \widehat{V}_{K,\mathcal{A}^-,\varepsilon}^{\text{Bayes},*}, s)$ to show the dependence on estimated probabilities and Bayesian value function. So the true Bayesian value function $V_{K,\mathcal{A}^-,\varepsilon}^{\text{Bayes},*}(s) = V(P(\boldsymbol{\sigma}), P(s'|s,a), V_{K,\mathcal{A}^-,\varepsilon}^{\text{Bayes},*}, s)$.

Hence, for any state $s \in \mathcal{S}$, the maximal estimation error can be decomposed by:

$$
\max_s |\widehat{V}_{K,\mathcal{A}^-,\varepsilon}^{\text{Bayes},*}(s) - V_{K,\mathcal{A}^-,\varepsilon}^{\text{Bayes},*}(s)|
$$
$$
= \max_s \left| V(\widehat{P}(\boldsymbol{\sigma}), \widehat{P}(s'|s,a), \widehat{V}_{K,\mathcal{A}^-,\varepsilon}^{\text{Bayes},*}, s) - V(P(\boldsymbol{\sigma}), P(s'|s,a), V_{K,\mathcal{A}^-,\varepsilon}^{\text{Bayes},*}, s) \right|
$$
$$
\leq \underbrace{\max_s \left| V(\widehat{P}(\boldsymbol{\sigma}), \widehat{P}(s'|s,a), \widehat{V}_{K,\mathcal{A}^-,\varepsilon}^{\text{Bayes},*}, s) - V(\widehat{P}(\boldsymbol{\sigma}), \widehat{P}(s'|s,a), V_{K,\mathcal{A}^-,\varepsilon}^{\text{Bayes},*}, s) \right|}_{T_{21}}
$$
$$
+ \underbrace{\max_s \left| V(\widehat{P}(\boldsymbol{\sigma}), \widehat{P}(s'|s,a), V_{K,\mathcal{A}^-,\varepsilon}^{\text{Bayes},*}, s) - V(P(\boldsymbol{\sigma}), \widehat{P}(s'|s,a), V_{K,\mathcal{A}^-,\varepsilon}^{\text{Bayes},*}, s) \right|}_{T_{22}}
$$
$$
+ \underbrace{\max_s \left| V(P(\boldsymbol{\sigma}), \widehat{P}(s'|s,a), V_{K,\mathcal{A}^-,\varepsilon}^{\text{Bayes},*}, s) - V(P(\boldsymbol{\sigma}), P(s'|s,a), V_{K,\mathcal{A}^-,\varepsilon}^{\text{Bayes},*}, s) \right|}_{T_{23}}.
$$

Here, $T_{21}$ captures the bias in the estimated Bayesian value function, $T_{22}$ quantifies the error from imperfect predictiondistribution estimation, and $T_{23}$ reflects the error in onestep transition kernel estimation.

The remaining hurdle is to bound the terms $T_{21}$, $T_{22}$ and $T_{23}$, respectively. Lemmas F.1, F.2 and F.3 provide the desired bounds as follows:

**Lemma F.1** (Proof in Appendix F.1). *For any state $s$, the term $T_{21}$ satisfies:*

$$
T_{21} \leq \gamma^K \max_s |\widehat{V}_{K,\mathcal{A}^-,\varepsilon}^{\text{Bayes},*}(s) - V_{K,\mathcal{A}^-,\varepsilon}^{\text{Bayes},*}(s)|. \tag{43}
$$

**Lemma F.2** (Proof in Appendix F.2). *With probability at least $1 - \delta$, the term $T_{22}$ satisfies:*

$$
T_{22} \leq \frac{1}{1-\gamma} \sqrt{\frac{\log(2|\mathcal{S}|/\delta)}{2N_2}}. \tag{44}
$$

**Lemma F.3** (Proof in Appendix F.3). *With probability at least $1 - \delta$, the term $T_{23}$ satisfies:*

$$
T_{23} \leq \frac{\gamma(1-\gamma^K)}{(1-\gamma)^2} \sqrt{\frac{\log(2K^2|\mathcal{S}|(|\mathcal{A}|-|\mathcal{A}^-|)/\delta)}{2N_1}}. \tag{45}
$$

Combining the results in Eq. (43), (44), (45) yields that, with probability at least $1 - \delta$

$$
\max_s \left| \widehat{V}_{K,\mathcal{A}^-,\varepsilon}^{\text{Bayes},*}(s) - V_{K,\mathcal{A}^-,\varepsilon}^{\text{Bayes},*}(s) \right| \leq \frac{\frac{1}{1-\gamma}\sqrt{\frac{\log(4|\mathcal{S}|/\delta)}{2N_2}} + \frac{\gamma(1-\gamma^K)}{(1-\gamma)^2}\sqrt{\frac{\log(1+4K^2|\mathcal{S}|(|\mathcal{A}|-|\mathcal{A}^-|)/\delta)}{2N_1}}}{(1-\gamma^K)}
$$
$$
\leq \frac{\sqrt{\frac{\log(4|\mathcal{S}|/\delta)}{2N_2}}}{(1-\gamma)(1-\gamma^K)} + \frac{\sqrt{\frac{\log(1+4K^2|\mathcal{S}|(|\mathcal{A}|-|\mathcal{A}^-|)/\delta)}{2N_1}}}{(1-\gamma)^2}.
$$

Letting $\frac{\sqrt{\frac{\log(4|\mathcal{S}|/\delta)}{2N_2}}}{(1-\gamma)(1-\gamma^K)} = \alpha\epsilon$ and $\frac{\sqrt{\frac{\log\left(1+4K^2|\mathcal{S}|(|\mathcal{A}|-|\mathcal{A}^-|)/\delta\right)}{2N_1}}}{(1-\gamma)^2} = (1-\alpha)\epsilon$ yields that:

$$N_1 = \frac{2\log\left(1+4K^2|\mathcal{S}|(|\mathcal{A}|-|\mathcal{A}^-|)/\delta\right)}{(1-\gamma)^4(1-\alpha)^2\epsilon^2}, \tag{46}$$

$$N_2 = \frac{2\log\left(4|\mathcal{S}|/\delta\right)}{(1-\gamma)^2(1-\gamma^K)^2\alpha^2\epsilon^2}. \tag{47}$$

Injecting Eq. (46) and (47) into Eq. (39) and combining the constants yields our result.

### F.1  Proof of Lemma F.1

For any fixed state $s$, the only difference between the two expressions in $T_{21}$ lies in their terminal Bayesian value function. Hence, by the $\gamma^K$ contraction property of the Bayesian Bellman operator, we immediately obtain:

$$\max_s \left| V(\widehat{P}(\boldsymbol{\sigma}), \widehat{P}(s'|s,a), \widehat{V}_{K,\mathcal{A}^-,\varepsilon}^{\mathrm{Bayes},*}, s) - V(\widehat{P}(\boldsymbol{\sigma}), \widehat{P}(s'|s,a), V_{K,\mathcal{A}^-,\varepsilon}^{\mathrm{Bayes},*}, s) \right|$$

$$= \max_{s_0} \left| \int_{\boldsymbol{\sigma}_{1:K}\in\mathcal{Q}_K} \widehat{P}(d\boldsymbol{\sigma}_{1:K}) \max_{\boldsymbol{a}} \left( \sum_{t=0}^{K-1} \gamma^t \left( \sum_{s_t} \widehat{P}(s_t|s,\boldsymbol{a}_{0:t-1},\boldsymbol{\sigma}_{1:t}) r(s_t,a_t) \right) \right. \right.$$

$$+ \gamma^K \sum_{s_K} \widehat{P}(s_K|s,\boldsymbol{a},\boldsymbol{\sigma}) \widehat{V}_{K,\mathcal{A}^-,\varepsilon}^{\mathrm{Bayes},*}(s_K)) \Bigg)$$

$$- \int_{\boldsymbol{\sigma}_{1:K}\in\mathcal{Q}_K} \widehat{P}(d\boldsymbol{\sigma}_{1:K}) \max_{\boldsymbol{a}} \left( \sum_{t=0}^{K-1} \gamma^t \left( \sum_{s_t} \widehat{P}(s_t|s,\boldsymbol{a}_{0:t-1},\boldsymbol{\sigma}_{1:t}) r(s_t,a_t) \right) \right.$$

$$\left. \left. + \gamma^K \sum_{s_K} \widehat{P}(s_K|s,\boldsymbol{a},\boldsymbol{\sigma}) V_{K,\mathcal{A}^-,\varepsilon}^{\mathrm{Bayes},*}(s_K)) \right) \right|$$

$$\leq \int_{\boldsymbol{\sigma}_{1:K}\in\mathcal{Q}_K} \widehat{P}(d\boldsymbol{\sigma}_{1:K}) \max_{s,\boldsymbol{a}} \left| \gamma^K \sum_{s_K} \widehat{P}(s_K|s,\boldsymbol{a},\boldsymbol{\sigma}) (V_{K,\mathcal{A}^-,\varepsilon}^{\mathrm{Bayes},*}(s_K) - V_{K,\mathcal{A}^-,\varepsilon}^{\mathrm{Bayes},*}(s_K)) \right|$$

$$\leq \gamma^K \max_s |\widehat{V}_{K,\mathcal{A}^-,\varepsilon}^{\mathrm{Bayes},*}(s) - V_{K,\mathcal{A}^-,\varepsilon}^{\mathrm{Bayes},*}(s)|.$$

This concludes our proof.

### F.2  Proof of Lemma F.2

We denote $X(\boldsymbol{\sigma},s)$ as the cumulative reward from state $s$ with $K$-step transition prediction $\boldsymbol{\sigma}$:

$$X(\boldsymbol{\sigma},s) = \max_{\boldsymbol{a}} \left( \sum_{t=0}^{K-1} \gamma^t \left( \sum_{s_t} \widehat{P}(s_t|s,\boldsymbol{a}_{0:t-1},\boldsymbol{\sigma}_{1:t}) r(s_t,a_t) \right) + \gamma^K \sum_{s_K} \widehat{P}(s_K|s,\boldsymbol{a},\boldsymbol{\sigma}) V_{K,\mathcal{A}^-,\varepsilon}^{\mathrm{Bayes},*}(s_K)) \right),$$

where $s_0 = s$. Note that, for any state $s$, $\widehat{P}(s_t|s,\boldsymbol{a}_{0:t-1},\boldsymbol{\sigma}_{1:t})$ and $V_{K,\mathcal{A}^-,\varepsilon}^{\mathrm{Bayes},*}(s_K)$, $X(\boldsymbol{\sigma},s)$ is bounded with $0 \leq X(\boldsymbol{\sigma},s) \leq \frac{1}{1-\gamma}$. Hence, for any state $s \in \mathcal{S}$, we have:

$$V(\widehat{P}(\boldsymbol{\sigma}),\widehat{P}(s'|s,a),V_{K,\mathcal{A}^-,\varepsilon}^{\mathrm{Bayes},*},s) - V(P(\boldsymbol{\sigma}),\widehat{P}(s'|s,a),V_{K,\mathcal{A}^-,\varepsilon}^{\mathrm{Bayes},*},s)$$

$$= \int_{\boldsymbol{\sigma}_{1:K}\in\mathcal{Q}_K} \widehat{P}(d\boldsymbol{\sigma}_{1:K}) X(\boldsymbol{\sigma},s) - \int_{\boldsymbol{\sigma}_{1:K}\in\mathcal{Q}_K} P(d\boldsymbol{\sigma}_{1:K}) X(\boldsymbol{\sigma},s)$$

$$= \int_{\boldsymbol{\sigma}_{1:K}\in\mathcal{Q}_K} (\widehat{P}(d\boldsymbol{\sigma}_{1:K}) - P(d\boldsymbol{\sigma}_{1:K})) X(\boldsymbol{\sigma},s). \tag{48}$$

To bound the distribution estimation error. We reformulate Eq. (48) into a finite-sample Hoeffding form. Specifically, we define $\overline{X}(s) = \int_{\boldsymbol{\sigma}_{1:K}\in\mathcal{Q}_K} P(d\boldsymbol{\sigma}_{1:K}) X(\boldsymbol{\sigma},s)$ and $X^i = X(\boldsymbol{\sigma}^i,s)$. Then, we

have:

$$\int_{\boldsymbol{\sigma}_{1:K}\in\mathcal{Q}_K}(\widehat{P}(d\boldsymbol{\sigma}_{1:K})-P(d\boldsymbol{\sigma}_{1:K}))X(\boldsymbol{\sigma},s)=\sum_{i=1}^{N_2}X^i(s)-\overline{X}(s),$$

where $\mathbb{E}_{\boldsymbol{\sigma}}(X^i(s))=\overline{X}(s)$ for all $s$. Applying the Hoeffding's inequality yields:

$$P\left(\left|\frac{1}{N_2}\sum_{i=1}^{N_2}X^i(s)-\overline{X}(s)\right|\geq\epsilon\right)\leq 2\exp\left(-\frac{2N_2\epsilon^2}{(\frac{1}{(1-\gamma)})^2}\right)=2\exp\left(-2(1-\gamma)^2N_2\epsilon^2\right),$$

where the first inequality holds because $0\leq X^i(s)\leq\frac{1}{(1-\gamma)}$ for all $i$. Letting the RHS probability term be $\frac{\delta}{|\mathcal{S}|}$, we apply the union bound across all different states $s\in\mathcal{S}$ yields:

$$T_{22}=\max_s\left|\sum_{i=1}^{N_2}X^i(s)-\overline{X}(s)\right|\leq\sqrt{\frac{\log\left(2|\mathcal{S}|/\delta\right)}{2(1-\gamma)^2N_2}}.$$

### F.3 Proof of Lemma F.3

This proof focuses on bounding how errors from inaccurate single-step transition kernels propagate to the long-term cumulative reward. The key idea is to decompose the total cumulative error into multiple time-dependent components and bound each individually.

**Step 1: Decompose the estimation error of $\widehat{P}(s_t|s,\boldsymbol{a}_{0:t-1},\boldsymbol{\sigma}_{1:t})$:**

For any estimation $\widehat{P}(s_t|s,\boldsymbol{a}_{0:t-1},\boldsymbol{\sigma}_{1:t})$, the term $T_{23}$ is upper bounded by:

$$
\begin{aligned}
T_{23}\leq\max_{s,\boldsymbol{\sigma}}&\left|\max_{\boldsymbol{a}}\left(\sum_{t=0}^{K-1}\gamma^t\left(\sum_{s_t}\widehat{P}(s_t|s,\boldsymbol{a}_{0:t-1},\boldsymbol{\sigma}_{1:t})r(s_t,a_t)\right)+\gamma^K\sum_{s_K}\widehat{P}(s_K|s,\boldsymbol{a},\boldsymbol{\sigma})V_{K,\mathcal{A}^-,\boldsymbol{\varepsilon}}^{\mathrm{Bayes},*}(s_K)\right)\right.\\
&\left.-\max_{\boldsymbol{a}}\left(\sum_{t=0}^{K-1}\gamma^t\left(\sum_{s_t}P(s_t|s,\boldsymbol{a}_{0:t-1},\boldsymbol{\sigma}_{1:t})r(s_t,a_t)\right)+\gamma^K\sum_{s_K}P(s_K|s,\boldsymbol{a},\boldsymbol{\sigma})V_{K,\mathcal{A}^-,\boldsymbol{\varepsilon}}^{\mathrm{Bayes},*}(s_K)\right)\right|\\
\leq\max_{s,\boldsymbol{\sigma},\boldsymbol{a}}&\left|\left(\sum_{t=0}^{K-1}\gamma^t\left(\sum_{s_t}\widehat{P}(s_t|s,\boldsymbol{a}_{0:t-1},\boldsymbol{\sigma}_{1:t})r(s_t,a_t)\right)+\gamma^K\sum_{s_K}\widehat{P}(s_K|s,\boldsymbol{a},\boldsymbol{\sigma})V_{K,\mathcal{A}^-,\boldsymbol{\varepsilon}}^{\mathrm{Bayes},*}(s_K)\right)\right.\\
&\left.-\left(\sum_{t=0}^{K-1}\gamma^t\left(\sum_{s_t}P(s_t|s,\boldsymbol{a}_{0:t-1},\boldsymbol{\sigma}_{1:t})r(s_t,a_t)\right)+\gamma^K\sum_{s_K}P(s_K|s,\boldsymbol{a},\boldsymbol{\sigma})V_{K,\mathcal{A}^-,\boldsymbol{\varepsilon}}^{\mathrm{Bayes},*}(s_K)\right)\right|\\
=\max_{s,\boldsymbol{\sigma},\boldsymbol{a}}&\left|\sum_{t=1}^{K-1}\gamma^t\left(\sum_{s_t}(\widehat{P}(s_t|s,\boldsymbol{a}_{0:t-1},\boldsymbol{\sigma}_{1:t})-P_t(s_t|s,\boldsymbol{a}_{0:t-1},\boldsymbol{\sigma}_{1:t}))r(s_t,a_t)\right)\right.\\
&\left.+\gamma^K\sum_{s_K}(\widehat{P}(s_K|s,\boldsymbol{a},\boldsymbol{\sigma})-P_K(s_K|s,\boldsymbol{a},\boldsymbol{\sigma}))V_{K,\mathcal{A}^-,\boldsymbol{\varepsilon}}^{\mathrm{Bayes},*}(s_K)\right|,\\
\leq\sum_{t=1}^{K-1}\gamma^t&\underbrace{\max_{s,\boldsymbol{\sigma},\boldsymbol{a}}\left|\sum_{s_t}(\widehat{P}(s_t|s,\boldsymbol{a}_{0:t-1},\boldsymbol{\sigma}_{1:t})-P_t(s_t|s,\boldsymbol{a}_{0:t-1},\boldsymbol{\sigma}_{1:t}))r(s_t,a_t)\right|}_{\Delta_t}\\
&+\gamma^K\underbrace{\max_{s,\boldsymbol{\sigma},\boldsymbol{a}}\left|\sum_{s_K}(\widehat{P}(s_K|s,\boldsymbol{a},\boldsymbol{\sigma})-P_K(s_K|s,\boldsymbol{a},\boldsymbol{\sigma}))V_{K,\mathcal{A}^-,\boldsymbol{\varepsilon}}^{\mathrm{Bayes},*}(s_K)\right|}_{\overline{\Delta}_K}.
\end{aligned}
$$

Now we have

$$T_{23}\leq\sum_{t=1}^{K-1}\gamma^t\Delta_t+\gamma^K\overline{\Delta}_K.\tag{49}$$

Specifically, for any $t$, initial state $s$, the action sequence $\boldsymbol{a}_{0:k-1}$ and prediction $\boldsymbol{\sigma}_{1:K}$, $\Delta_t$ can be represented as the following product form:

$$\Delta_t = (P_0 \cdot \widehat{P}_{0:1} \cdot \ldots \cdot \widehat{P}_{t-1:t} - P_0 \cdot P_{0:1} \cdot \ldots \cdot P_{t-1:t}) \cdot r^s_{\max}, \tag{50}$$

where vector $P_0 \in \mathbb{R}^{1 \times |\mathcal{S}|}$ denotes the initial state $s$ with the $s$-th entry equals 1 and the other entries equal 0, $r^s_{\max} \in \mathbb{R}^{|\mathcal{S}| \times 1}$ denotes the vector of reward with the $s$-th entry equals $\max_a r(s, a)$. Matrices $\widehat{P}_{i:i+1} \in \mathbb{R}^{|\mathcal{S}| \times |\mathcal{S}|}$ and $P_{i:i+1} \in \mathbb{R}^{|\mathcal{S}| \times |\mathcal{S}|}$ denote the estimated and real state transition probabilities from state $s_i$ to state $s_{i+1}$ with given action $a_i$.

Hence, $\Delta_t$ can be further decomposed by one-step errors as follows:

$$\begin{aligned}
\Delta_t &= (P_0 \cdot \widehat{P}_{0:1} \cdot \ldots \cdot \widehat{P}_{t-1:t} - P_0 \cdot P_{0:1} \cdot \ldots \cdot P_{t-1:t}) \cdot r^s_{\max} \\
&= P_0 \cdot (\widehat{P}_{0:1} \cdot \ldots \cdot \widehat{P}_{t-1:t} - P_0 \cdot P_{0:1} \cdot \ldots \cdot P_{t-1:t}) \cdot r^s_{\max} \\
&= \sum_{i=1}^{t} \left( P_0 \cdot \left( \prod_{j=0}^{i-1} \widehat{P}_{j:j+1} \prod_{k=i}^{t-1} P_{k,k+1} - \prod_{j=0}^{i-2} \widehat{P}_{j:j+1} \prod_{k=i-1}^{t-1} P_{k,k+1} \right) \cdot r^s_{\max} \right) \\
&= \sum_{i=1}^{t} \underbrace{\left( P_0 \cdot \left( \prod_{j=0}^{i-2} \widehat{P}_{j:j+1} (\widehat{P}_{i-1:i} - P_{i-1:i}) \prod_{k=i}^{t-1} P_{k,k+1} \right) \cdot r^s_{\max} \right)}_{\Delta_{t,i}}. \tag{51}
\end{aligned}$$

Similarly, the terminal value error $\overline{\Delta}_K$ can be decomposed as:

$$\begin{aligned}
\overline{\Delta}_K &= (P_0 \cdot \widehat{P}_{0:1} \cdot \ldots \cdot \widehat{P}_{t-1:t} - P_0 \cdot P_{0:1} \cdot \ldots \cdot P_{t-1:t}) \cdot \widehat{V}^{\text{Bayes},*}_{K,\mathcal{A}^-,\varepsilon}(s^K) \\
&= \sum_{i=1}^{t} \underbrace{\left( P_0 \cdot \left( \prod_{j=0}^{i-2} \widehat{P}_{j:j+1} \left( \widehat{P}_{i-1:i} - P_{i-1:i} \right) \prod_{k=i}^{t-1} P_{k,k+1} \right) \cdot \widehat{V}^{\text{Bayes},*}_{K,\mathcal{A}^-,\varepsilon} \right)}_{\overline{\Delta}_{K,i}}. \tag{52}
\end{aligned}$$

Combining Eq. (51), (52) with Eq. (49), (50), we can decompose total error $T_{23}$ into:

$$T_{23} \le \sum_{t=1}^{K-1} \gamma^t \sum_{i=1}^{t} \Delta_{t,i} + \gamma^K \sum_{i=1}^{t} \overline{\Delta}_{K,i}. \tag{53}$$

**Step 2: Bound individual error terms $\Delta_{t,i}$ and $\overline{\Delta}_K$:**

We can show the concentration behavior of $|\Delta_{t,i}|$ as follows

$$\begin{aligned}
|\Delta_{t,i}| &\le \left| \left( P_0 \cdot \left( \prod_{j=0}^{i-2} \widehat{P}_{j:j+1} (\widehat{P}_{i-1:i} - P_{i-1:i}) \prod_{k=i}^{t-1} P_{k,k+1} \right) \cdot r^s_{\max} \right) \right| \\
&\le \left| P_0 \cdot \left( \prod_{j=0}^{i-2} \widehat{P}_{j:j+1} \right) \right| \left| \left( \widehat{P}_{i-1:i} - P_{i-1:i} \right) \left( \prod_{k=i}^{t-1} P_{k,k+1} \cdot r^s_{\max} \right) \right| \\
&\le \left| \left( \widehat{P}_{i-1:i} - P_{i-1:i} \right) \left( \prod_{k=i}^{t-1} P_{k,k+1} \cdot r^s_{\max} \right) \right| \\
&= \left| \left( \widehat{P}_{i-1:i} - P_{i-1:i} \right) r^{s_t,s}_{\max} \right|,
\end{aligned}$$

where $r^{s_t,s}_{\max} = \prod_{k=i}^{t-1} P_{k,k+1} \cdot r^s_{\max} \in \mathbb{R}^{|\mathcal{S}|}$ denotes the expected reward vector from state $s_t$, which satisfies $|r^{s_t,s}_{\max}|_\infty \le 1$ for any $t$.

With $N_1$ samples on each substate-action pair $(s, a)$, we can directly bound $\left( \widehat{P}_{i-1:i} - P_{i-1:i} \right) r^{s_t,s}_{\max}$ by the Hoeffding's inequality:

$$P\left( |\Delta_{t,i}| \ge \epsilon \right) \le 2 \exp\left( -2N_1 \epsilon^2 \right),$$

We need to ensure $|\Delta_{t,i}| \ge \epsilon$ for any $(s, a)$ with $s \in \mathcal{S}$ and $a \in \mathcal{A} \backslash \mathcal{A}^-$, thus, we take $2 \exp\left( -\frac{2\epsilon^2}{N_1} \right) = \frac{\delta}{|\mathcal{S}|(|\mathcal{A}|-|\mathcal{A}^-|)}$ and yields that, with probability at least $1 - \delta$,

$$|\Delta_{t,i}| \le \sqrt{\frac{\log\left( 2|\mathcal{S}|(|\mathcal{A}| - |\mathcal{A}^-|)/\delta \right)}{2N_1}}, \forall t, i. \tag{54}$$

Following exactly the same routine, since $\widehat{V}_{K,\mathcal{A}^-,\varepsilon}^{\text{Bayes},*}(s) \in [0, \frac{1}{1-\gamma}]$, we have that, with probability at least $1 - \delta$, $\overline{\Delta}_{K,i}$ satisfies:

$$|\overline{\Delta}_{K,i}| \leq \frac{1}{1-\gamma}\sqrt{\frac{\log\left(2|\mathcal{S}|(|\mathcal{A}| - |\mathcal{A}^-|)/\delta\right)}{2N_1}}, \forall t, i. \tag{55}$$

**Step 3: Combine the pieces:** Now it is enough to bound $\Delta$. Let the probability $\delta$ in Eq. (54)-(55) be $\frac{\delta}{K^2}$, we have that, with probability at least $1 - \delta$,

$$
\begin{aligned}
|\Delta| &\leq \sum_{t=1}^{K-1} \gamma^t \sum_{i=1}^{t} |\Delta_{t,i}| + \gamma^K \sum_{i=1}^{t} |\overline{\Delta}_{K,i}| \\
&\leq \sum_{t=1}^{K-1} \gamma^t \sum_{i=1}^{t} \sqrt{\frac{\log\left(2K^2|\mathcal{S}|(|\mathcal{A}| - |\mathcal{A}^-|)/\delta\right)}{2N_1}} \\
&\quad + \gamma^K \sum_{i=1}^{t} \frac{1}{1-\gamma}\sqrt{\frac{\log\left(2K^2|\mathcal{S}|(|\mathcal{A}| - |\mathcal{A}^-|)/\delta\right)}{2N_1}} \\
&\leq \sqrt{\frac{\log\left(2K^2|\mathcal{S}|(|\mathcal{A}| - |\mathcal{A}^-|)/\delta\right)}{2N_1}}\left(\sum_{t=1}^{K-1} t\gamma^t + \frac{K\gamma^K}{1-\gamma}\right) \\
&\leq \frac{\gamma(1-\gamma^K)}{(1-\gamma)^2}\sqrt{\frac{\log\left(2K^2|\mathcal{S}|(|\mathcal{A}| - |\mathcal{A}^-|)/\delta\right)}{2N_1}}.
\end{aligned}
$$

This concludes our proof.

# G   Detailed Model of Wind-Farm Storage Control

We consider a windfarm operator that must deliver a precommitted power schedule to the grid while managing the uncertainty of realtime wind generation. To avoid costly imbalances, the farm uses a battery storage system: when actual output exceeds the commitment, excess energy is stored; when output falls short, the storage discharges to make up the difference. The operators goal is to minimize cumulative penalty costs for over or underdelivery by choosing charge/discharge actions based on observed prices, the current stateofcharge, and shortterm forecasts of wind generation.

Below, we formalize this sequential decision problem as an MDP.

## G.1   MDP Formulation for WindFarm Storage Control

We model sequential storage control as an MDP $(\mathcal{S}, \mathcal{A}, \mathcal{P}, r, \gamma)$ with:

- **State**: $s_t = (p_t, \Delta_t, \text{SoC}_t)$, where $p_t$ is the (identical) penalty price, $\Delta_t$ is the generation mismatch between generated wind power $w_t$ and required wind power $\widehat{w}_t$, and $\text{SoC}_t$ is the storages state of charge.
- **Action**: $a_t = (v_t^+, v_t^-)$, denoting charge/discharge amounts subject to $v_t^+ v_t^- = 0$, capacity, and generation constraints.
- **Transition**: the state transition probabilities satisfy:

$$P(s_{t+1} \mid s_t, a_t) = P(p_{t+1} \mid p_t)\, P(\Delta_{t+1} \mid \Delta_t)\, \mathbf{1}\{\text{SoC}_{t+1} = \text{SoC}_t + \eta^+ v_t^+ - \eta^- v_t^-\}.$$

- **Reward**: the reward function is the negative penalty defined as:

$$r(s_t, a_t) = -\left[p_t \max(\Delta_t - v_t^+, 0) + p_t \max(v_t^- - \Delta_t, 0)\right].$$

The dynamics of the storage control problem is visualized in Figure 3.

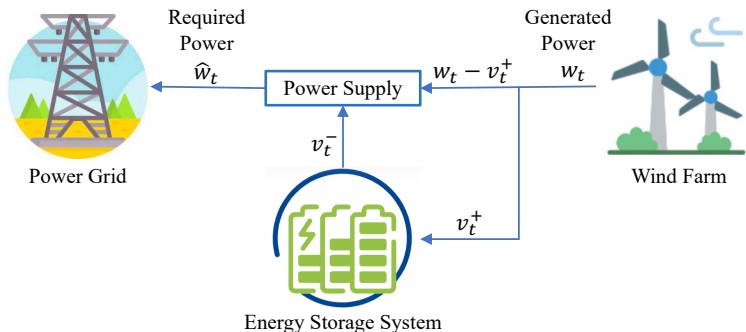

Figure 3: Wind Farm Storage Control

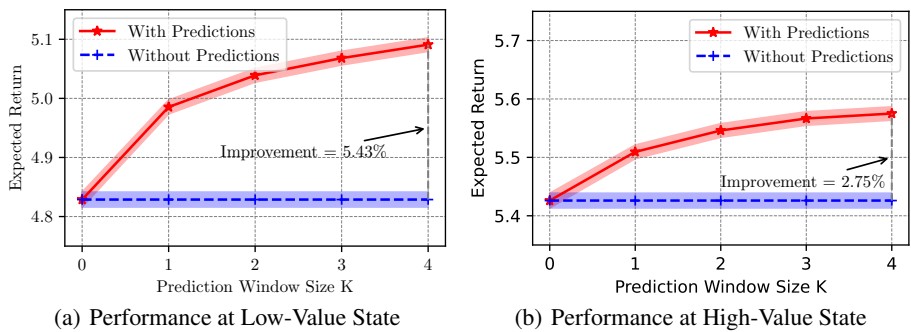

(a) Performance at Low-Value State

(b) Performance at High-Value State

Figure 4: Prediction Helps More in Low-Value States. (a): The expected return at a low-value state increases significantly with prediction horizon $K$ (b): The return at a high-value state also improves with $K$, but the marginal gain is much smaller.

## G.2 Parameter Settings

In the numerical study, we utilized the California aggregate wind power generation dataset from CAISO [60] containing predicted and real wind power generation data with a 5-minute resolution spanning from January 2020 to December 2020. The penalty price equals the average electricity price of CASIO [60] with the matching resolution and periods. We set $C = 10$ kWh, $\gamma = 0.95$. The discretization levels of $p$, $\Delta w$ and $SoC$ are set to be 10, 10, 21, respectively. The action set includes 9 discretized choices ranging from charging 2 KWh to discharging 2 KWh. The other parameters follow [61].

## H  Additional Experiments

In this section, we conduct experiments to validate our theoretical findings. In particular, we demonstrate the advantage of Algorithm 1 for MDPs with transition predictions, compared with model-based RL for classical MDPs. For each experiment, we randomly generate 20 MDPs with $|\mathcal{S}| = 10$ and $|\mathcal{A}| = 5$, and present the average performance of both approaches.

We first verify how predictions improve expected returns over standard MDPs, as predicted by our theoretical analysis in Section 4.2. Specifically, we examine two representative states—one with the lowest value and one with the highest value under the standard MDP value function. Figure 4(a) illustrates how incorporating transition predictions enhances the value functions of these two states. Even with a short prediction horizon $K = 1$, we observe notable improvements over the MDP baseline. As $K$ increases, the improvements also increase and tend to converge, which aligns with our theoretical findings. Interestingly, we find that low-value states benefit more from predictions than high-value states. In particular, for the low-value state, the expected return improves by $5.43\%$ with $K = 4$. In contrast, the corresponding gain for the high-value state is $2.75\%$. This is intuitive because predictions help guide the agent toward transitions that reach higher-value regions, offering more substantial gains for states with lower initial value.

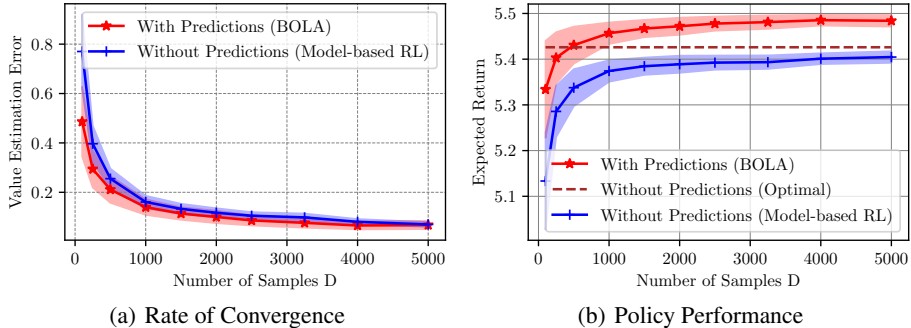

(a) Rate of Convergence  (b) Policy Performance

Figure 5: Sample Efficiency of Learning with Predictions. (a): Convergence of (Bayesian) value function estimation error; (b): Policy performance measured by expected return. BOLA with predictions achieves similar convergence and consistently higher performance across varying sample sizes compared to classical model-based RL.

Figure 5 compares the learning efficiency of BOLA with $K = 1$ and standard model-based RL. In Figure 5(a), BOLA exhibits a faster decay in the value estimation error with fewer environment samples, indicating its sample efficiency is no worse than vanilla MDPs even with $K = 1$. In Figure 5(b), we observe that BOLA consistently outperforms the model-based RL baseline. Notably, once the number of samples exceeds a small threshold $500$, the policy learned via BOLA yields higher expected return than what is maximally achievable by any MDP policy without predictive information. This highlights the fundamental advantage of incorporating predictions, which enables agents to surpass the conventional performance ceiling imposed by standard MDP frameworks.

# I   Extension to Splitable State Modeling

Our model can be extended to Markov Decision Processes (MDPs) with *splittable states*, which naturally generalize to settings with predictable trajectories. A key feature of this extension is the decomposition of each state $s \in \mathcal{S}$ into two independent components, represented as a pair $s = (s^m, s^d)$, where $s^m \in \mathcal{S}^m$ is the *Markovian substate*, and $s^d \in \mathcal{S}^d$ is the *dependent substate*, with $\mathcal{S} = \mathcal{S}^m \times \mathcal{S}^d$.

This modeling approach provides a natural way to incorporate exogenous or independently fore-castable time seriessuch as demand, weather, or price signalsinto the decision-making process. Specifically, these predictable sequences can be encoded in the Markovian substate $s^m$, allowing the agent to plan adaptively using trajectory-level predictions without enlarging the core Markov state space. The two substates are formally defined below.

**Markovian Substates.** The first type of substates, denoted by $s^m$, is used to capture externally evolving states that do not depend on the agent's action, with several important real-world examples to be discussed in Section I.1. State transitions with respect to $s^m$ are Markovian such that they only depend on the previous substate. Formally, its transition kernel satisfies

$$P(s^m_{t+1} \mid s_t, a_t) = P(s^m_{t+1} | s^m_t), \quad \forall s_{t+1}, s_t \in \mathcal{S}, \text{ and } a \in \mathcal{A}. \tag{56}$$

**Dependent Substates.** The transition of this substate, denoted by $s^d$, depends on both past substate and action (like in classical MDPs). The state transitions with respect to $s$ are

$$P\left(s^d_{t+1} \mid s_t, a_t\right) = P\left(s^d_{t+1} \mid s^d_t, a\right), \quad \forall s_{t+1}, s_t \in \mathcal{S}, \text{ and } a \in \mathcal{A}. \tag{57}$$

The Markovian substate and the dependent substate are assumed to have independent transitions, i.e., for any $s_t = (s^m_t, s^d_t) \in \mathcal{S}$, $a \in \mathcal{A}$, the overall transition probability $P(s_{t+1} \mid s_t, a_t)$ satisfies $P(s_{t+1} \mid s_t, a_t) = P(s^m_{t+1} \mid s^m_t)P(s^d_{t+1} \mid s^d_t, a_t)^2$. Unlike Markovian substates, a dependent substate depends on both past action and substate, making it harder to predict. Next, we introduce the prediction model of the prediction-augmented MDPs considered in this work.

---

[2]Without the predictive model $\mathcal{P}$, the MDP model with Markovian and dependent substates is essentially a special case of the factored MDP [62].

Let $\mathcal{P} = (K, \mathcal{A}^-, \boldsymbol{\sigma})$ denote the prediction model that provides the predictions to the future states, where $K$ is the length of the prediction window, $\mathcal{A}^- \subseteq \mathcal{A}$ is a predictable action set, and $\boldsymbol{\sigma}$ denotes the transition prediction. Specifically, given $t = 0, K, 2K, \ldots$, let $s_t = (s_t^m, s_t^d)$ be the current state of the environment. Before taking actions, the agent receives a probablistic prediction of the transition $\boldsymbol{\sigma} = (\sigma_1, \sigma_2, \cdots, \sigma_K)$ for the next $K$ steps, where different $\sigma_k$'s are independent and sampled from an unknown probability distribution $P(\sigma_k)$. For each $k \in \{1, 2, \cdots, K\}$, $\sigma_k$ captures the transitions from state $s_{t+k-1}$ to state $s_{t+k}$, and is of the form $\sigma_k = (\sigma_k^m, \sigma_k^d)$, where $\sigma_k^m \in [0, 1]^{|\mathcal{S}| \times |\mathcal{S}|}$ is an $|\mathcal{S}| \times |\mathcal{S}|$-dimensional matrix representing the transition prediction of the Markovian substate and $\sigma_k^d \in [0, 1]^{|\mathcal{S}||\mathcal{A}^-| \times |\mathcal{S}|}$ is an $|\mathcal{S}||\mathcal{A}^-|$ by $|\mathcal{S}|$ matrix representing the transition prediction of the dependent substate. Given a prediction $\sigma = (\sigma^m, \sigma^d)$, the transition probabilities satisfy the following.

**Fully Predictable Markovian Substates.** For Markovian substates, we have

$$P\left(s_{t+1}^m | s_t^m, \sigma_{t+1}\right) = \sigma_{t+1}^m\left(s_t^m, s_{t+1}^m\right), \tag{58}$$

where $\sigma^m(s_t^m, s_{t+1}^m)$ is the $(s_t^m, s_{t+1}^m)$-th entry of the matrix $\sigma^m$.

**Partially Predictable Dependent Substates.** We consider a general setting that allows for partially predictable states and actions. In particular, given a set of predictable actions $\mathcal{A}^-$, we have

$$P\left(s_{t+1}^d | s_t^d, a_t, \sigma_{t+1}^d\right) = \begin{cases} \sigma_{t+1}^d((s_t^d, a_t), s_{t+1}^d), & \text{if } a \in \mathcal{A}^-, \\ P\left(s_{t+1}^d | s_t^d, a_t\right), & \text{if } a \notin \mathcal{A}^-. \end{cases} \tag{59}$$

where $\sigma_{t+1}^d((s_t^d, a_t), s_{t+1}^d)$ denotes the entry located in the $(s_t^d, a_t)$-th row and the $s_{t+1}^d$-th column of the matrix $\sigma_{t+1}^d$.

## I.1   Illustrative Examples

Extending classic MDPs, the prediction-augmented MDP model introduced in Section 2.1 naturally fits real-world scenarios with Markovian states. Examples include stock prices in stock market trading [8], outdoor temperatures in building thermal control [63], wind speeds in unmanned aerial vehicle (UAV) control [64], electricity prices in storage control [65], and grid electricity demands in power system economic dispatch [66].

Table 1: Real-world examples that instantiate the prediction-augmented MDP model (see Section 2.1).

| | Action | Markovian Substate _predictable_ | Dependent Substate _partially predictable_ |
|---|---|---|---|
| Stock Investment | Buy/sell stocks | Stock price | N/A |
| VPP Operation | Energy consumption | Renewable generation | Electricity price |
| Building HVAC Control | Heating/cooling | N/A | Indoor/outside temperature |
| Storage Control | Charge/discharge | Energy mismatch | Battery SoC |
| UAV Control | DC motor force | Wind direction/speed | UAV position/attitude |

**Stock Investment.** Consider the stock investment problem for a retail investor [8]. The market stock prices are action-independent unknown time series if the trading volume is high. With relatively accurate stock price predictions, the revenue from the investment can be significantly improved.

**Virtual Power Plant operation.** Another example is the virtual power plant (VPP) operation problem [67], where the agent sequentially decides the energy consumption of a large-scale VPP to minimize total electricity costs. In this scenario, the renewable generation within the VPP depends solely on its previous state and can be effectively predicted. Conversely, the real-time electricity price in the electricity market depends on both its previous state and the VPP's energy consumption actions. The electricity price is partially predictable: when the VPP's energy consumption is low, its impact on market prices is minimal and predictable. However, when energy consumption is very high, the VPP becomes a market price-maker, causing market prices to fluctuate wildly and become unpredictable.

**Building HVAC Control.** Besides, many online decision-making problems related to sustainability exhibit predictable structures aligning with our model. For example, the control of heating, ventilation, and air conditioning (HVAC) systems relies on temperature predictions [68], battery storage

management depends on energy predictions [69]. Similar predictable components exist for the task of controlling battery storage systems.

In Table 1, we summarize the key features of real-world scenarios with Markovian substates and partially predictable dependent substates, which present challenges for modeling with classic MDPs.

