# OpenReview forum: "Reinforcement Learning with Imperfect Transition Predictions: A Bellman-Jensen Approach"
_NeurIPS.cc/2025/Conference — NeurIPS 2025 spotlight_

### Official Review · Reviewer_jBsU · 2025-07-02

**Clarity:** 3
**Significance:** 2
**Originality:** 3
**Rating:** 5
**Confidence:** 4

**Summary:**

This paper studies a reinforcement learning setting where the agent has access to an oracle of imperfect predictions. The text introduces the framework of Bayesian value functions and the prediction-aware optimal policy as well as a corresponding Bellman operator. Then, it provides a theoretical analysis of the effects of perfect predictions before a tabular algorithm BOLA for prediction-aware RL is presented. A theoretical analysis on the sample complexity is provided as well as a set of numerical studies.

**Questions:**

Q1: In the numerical study, can you elaborate on what happens in the case of no predictions? Does the Bayes Bellman operator not capture the regular Bellman operator?

Q2: Can you elaborate why the sample complexity of the algorithm would not have an O(|S||A|) dependence?

**Ethical Concerns:**

["NO or VERY MINOR ethics concerns only"]

**Final Justification:**

This paper presents an interesting view on combining RL with forecasts and I think the community will likely benefit from learning about this contribution.

**Limitations:**

The limitations are outlined in section 7, however I do think it would be beneficiall to add a discussion of the oracle assumption and what challenges it adds.

**Quality:**

3

**Strengths And Weaknesses:**

**Strengths**

**Motivation**
* Finding new frameworks that allow us to leverage future forecasts and predictions is quite relevant in current times where foundational models exist. As such, I find this a very interesting problem to study.

**Clarity**
* The text is well written, well structured and easy to follow
* The theoretical statements have in-depth intuitive explanations that make it easy to understand the relevant parts of the resulting bounds.
* The structure of results has a nice build-up that starts with intuition relating the framework to traditional MDPs, to studying perfect predictions before the final result is presented. This makes is quite easy to follow what results are being presented and why.

**Related Work**
* I’m not very familiar with the Bayesian side of RL but the treatment of related work is relatively extensive

**Novelty**
* I’m not aware of any work that attempts to provide similar results and I believe this may be an interesting novel contribution for the community.

**Claims and Evidence**
* The first claim is that a new low-dimensional value Bayesian value function that leverages multi-step predictions is introduced which is supported by section 3.
* The second claim is the novel decomposition into local Jensen gaps that allows to learn a policy in the proposed setting without much overhead. This claim is also
* The claim that BOLA circumvents the exponential blow-up from state augmentation is supported by the algorithms sample complexity.

**Experimental Design and Analyses**
* The experimental study highlights that the theoretical results hold in practice.

__________
**Neutral Points**

**Minor points**
* It might make sense for clarity purposes to rename the sub-gaussian variable since it shares notation with the predictions.
__________

**Weaknesses**

**Experimental Design and Analyses**

**Claims and Evidence**
* I’m a bit confused about the sample complexity D1. D1 is a function of |S|(|A| - |A-|) but also from what the text states |S||A|. The classical O(|S||A|) state dependence is still in D1. Even if the first term vanishes, the second term still exists. Thus I’m unsure if the claim that BOLA provides better sample complexity guarantees than regular model-based RL algorithms is correct. I will ask for clarification. Maybe this is meant to say there is no epsilon scaling on that part of the dependence?

**Theoretical Results**
* A fundamental question is whether or not these algorithms provide any benefits over traditional RL algorithms. I think this answer could be a bit stronger. In the end, the upper bound in the worst case must still depend on the state-action space size which in practice is quite limiting.
* The analysis and limitations ignore the cost of obtaining the prediction oracle. In the worst case, it seems one may have to learn a full model of the transition dynamics to obtain such a model before the algorithm can be run. In this case, it seems this type of algorithm would be more expensive than a traditional RL algorithm.

__________

**Summary**

Overall, I think this is a nice paper. While there are some drawbacks to the theoretical assumptions, I understand that this is a first result on this type of algorithm. These initial results follow the structure of many initial results in the RL community that later led to interesting practical findings. I’m a bit worried about the sample complexity statement but if my questions can be answered, I am happy to recommend acceptance.

---

> ### Author Rebuttal · Authors · 2025-07-30
>
> Thank you for the thoughtful and encouraging review. We appreciate your positive comments on the motivation, clarity, novelty, and theoretical contributions of the work. Below, we address your questions in detail.
>
> **Q1. I’m a bit confused about the sample complexity D1. D1 is a function of |S|(|A| - |A-|) but also from what the text states |S||A|. The classical O(|S| |A|) state dependence is still in D1. Even if the first term vanishes, the second term still exists. Thus I’m unsure if the claim that BOLA provides better sample complexity guarantees than regular model-based RL algorithms is correct. I will ask for clarification. Maybe this is meant to say there is no epsilon scaling on that part of the dependence?**
>
>
> **Response:** Thank you for pointing this out, and we appreciate the opportunity to clarify. You are absolutely right: our claim refers specifically to the $\epsilon$ scaling in the sample complexity. The classical minimax sample complexity for model-based RL is $\mathcal{O}(|S||A|\epsilon^{-2})$ [1], with model-free methods typically requiring even more samples. In contrast, our term $D_1$ has a lower complexity $\mathcal{O}(|S|(|A| - |A^{-}|)\epsilon^{-2} + |S||A|)$. The key distinction is that only the first term in $D_1$ depends on $\epsilon^{-2}$, while the second term, $\mathcal{O}(|S||A|)$, does not scale with $\epsilon$ and is often negligible.
> We will make this clarification explicit in the revision to avoid any confusion.
>
> >[1] Azar, M. G., Munos, R., & Kappen, H. (2012). On the Sample Complexity of Reinforcement Learning with a Generative Model. In International Conference on Machine Learning.
>
> **Q2. A fundamental question is whether or not these algorithms provide any benefits over traditional RL algorithms. I think this answer could be a bit stronger. In the end, the upper bound in the worst case must still depend on the state-action space size which in practice is quite limiting.**
>
> **Response:**  Yes, our algorithm can be more sample-efficient, especially when predictions are available (as discussed in our response to Q1).
>
> * **Theoretically, our sample complexity is $\mathcal{O}(|\mathcal{S}|(|\mathcal{A}| - |\mathcal{A}^-|)\epsilon^{-2})$, which is lower than the minimax sample complexity $\mathcal{O}(|\mathcal{S}||\mathcal{A}|\epsilon^{-2})$ in tabular MDPs [1]**. Intuitively, for actions $a \in \mathcal{A}^-$, transition dynamics are directly obtained from predictions $\sigma$, so increasing $|\mathcal{A}^-|$ leads to improved efficiency. The $|\mathcal{S}||\mathcal{A}|$ scaling can not be avoided in statistical theory (because we at least need to sample all the reward function $r(s,a)$), but our framework can avoid the $|\mathcal{S}||\mathcal{A}|\epsilon^{-2}$ scaling of standard MDPs.
>
> >[1] Azar, M. G., Munos, R., & Kappen, H. (2012). On the Sample Complexity of Reinforcement Learning with a Generative Model. In International Conference on Machine Learning.
>
> * **Empirically, our framework scales naturally to large or continuous state spaces using function approximation, similar to how standard MDPs overcome the sample complexity dependence on state-action space size.** The key idea is to approximate the Bayesian value function $V^{\text{Bayes}}(s)$ using function approximation (e.g., neural networks), trained via fitted value iteration. This is feasible because $V^{\text{Bayes}}(s)$ shares the same dimensionality and Bellman structure as the standard value function, with the only difference being the use of prediction $\sigma$. Once $V^{\text{Bayes}}(s)$ is learned, the optimal policy can be extracted via standard planning techniques.  We provide an example of realizing the algorithm in our setting as follows:
>
>
> Consider the case where the forecast horizon $K=1$. Given a dataset of transitions $\mathcal{D} = {(s_i, a_i, r_i, s_i')}$ and a dataset of one-step predictions $\mathcal{D}\sigma$, we aim to learn a Bayesian value function $\hat{V}^{\text{Bayes}}_\theta(s)$ using fitted value iteration:
>
> * Step1: Sample a batch of states $s$ from $\mathcal{D}$
> * Step 2: For each $s$, compute the Bellman target:
> $y(s) = \mathbb{E}{\sigma \sim \mathcal{D}\sigma} \left[ \max_{a \in \mathcal{A}} \left( r(s, a) + \gamma \cdot \frac{1}{M} \sum_{m=1}^M V_\theta(s^{(m,\sigma)}) \right) \right]$,
>
> where $s^{(m,\sigma)} \sim \hat{P}\phi(\cdot \mid s, a, \sigma)$ are next states sampled from a learned transition model $\hat{P}\phi$ using forecast $\sigma$.
> * Step 3: Update parameter $\theta$ of the value function by minimizing the squared error $\min_\theta \sum_{s \in \text{batch}} \left(V_\theta(s) - y(s)\right)^2$
> * Step 4: Repeat from Steps 2-3 until convergence.
>
>
>
>
> **Q3. The analysis and limitations ignore the cost of obtaining the prediction oracle. In the worst case, it seems one may have to learn a full model of the transition dynamics to obtain such a model before the algorithm can be run. In this case, it seems this type of algorithm would be more expensive than a traditional RL algorithm.**
>
>
> **Response:**  Thank you for raising this important point. In our analysis, we do not assume that prediction oracles are free; rather, we treat querying the oracle as incurring the same per-sample cost as interacting with the environment (e.g., one sample for each $\sigma$).
>
> * In many practical settings, such as electricity markets, weather systems, or queueing networks, forecast information is externally available through domain-specific models and does not need to be learned by the RL agent. Moreover, generating such forecasts is often much less costly than estimating full transition dynamics via trial-based exploration. For instance, in a discrete-time M/M/1 queue, forecasting only the arrival probability $\lambda$ and service probability $\mu$ is sufficient to compute all transition probabilities analytically, such as $P(s_{t+1} = j+1 \mid s_t = j) = \lambda(1 - \mu)$ (i.e., an arrival occurs but no service).
>
> * Our framework is also designed to flexibly support **partial prediction** (see Section 2.1), where only a subset of actions $\mathcal{A}^-$ are associated with transition forecasts. In this case, the agent only needs to learn the dynamics for the remaining actions $\mathcal{A} \setminus \mathcal{A}^-$, and we provide corresponding sample complexity guarantees.
>
> * Even under the most conservative assumption where the prediction $\sigma$ must be queried for each transition entry (e.g., $|S||\mathcal{A}^-|$ samples for each $\sigma$), the total cost including both oracle queries ($D_2 = \mathcal{O}(|\mathcal{S}||\mathcal{A}^-|\epsilon^{-2})$) and environment samples ($D_1 = \mathcal{O}(|\mathcal{S}|(|\mathcal{A}| - |\mathcal{A}^-|)\epsilon^{-2})$) still matches the standard minimax sample complexity of model-based RL: $\mathcal{O}(|\mathcal{S}||\mathcal{A}|\epsilon^{-2})$. Thus, our framework is never worse than classical MDPs in theory, and it can provide substantial benefits when more forecasts are available in practice.
>
>
>
> **Minor: It might make sense for clarity purposes to rename the sub-gaussian variable since it shares notation with the predictions.**
>
> **Response:**  Thanks for the suggestion, we will change the notation for clarity.
>
>
> **Again, thank you for your thoughtful review. We would be happy to further clarify any points if there are remaining questions during the discussion.**

---

> ### Comment · Reviewer_jBsU · 2025-08-03
>
> Dear authors,
>
> thank you for clarifying my point about the state-action dependence. This was really my main concern that must be addressed in a final version of this paper as the claim as stated in the manuscript right now is not correct. I trust the authors will faithfully do so and am willing to raise my score to accept. I think this paper is an interesting contribution and I enjoyed learning something from it. Given that I believe I understood this point correctly in the first place, I raised my confidence to 4.
>
> Now to the other points that really in my eyes don't really require discussion and should be seen as feedback. It simply might make sense to think about these and decide whether or not it is warranted to highlight them as limitations in some form of discussion.
>
> The statement "our algorithm can be more sample efficient" misses the point I am making in my comment. One of the biggest challenges in empirical RL is dealing with large or infinitely sized state-action spaces. If our cost depends on the state-action space it is likely infeasible in practice irrespective of any epsilon dependence. This is a limitation of this approach. I understand that the proposed Bellman operator can potentially be iterated on using function approximation. Whether or not that is in fact more sample efficient is not demonstrated in this work and it is out of scope.
>
> Secondly, I did not say that the paper argues that oracles are free. I said it ignores the fact that one needs to *have* an oracle.  If it is easy to obtain an oracle, and querying an oracle is only computationally slightly more expensive, practitioners would likely simply run the computer for longer. In such cases, one might even argue that the cost of executing the oracle is lower than obtaining an actual data sample. That is the promise of model-based RL approaches after all. Yet in many interesting scenarios one needs to first learn an oracle from data. In practice, one could attempt to learn an oracle from data using neural networks for instance. Whether or not learning a forecaster and then applying an approach such as this is more or less useful that straight-up learning the standard value function is unclear to me. Again, I don't expect this to be answered in this manuscript.

---

> > ### Author Response · Authors · 2025-08-05
> >
> > Thank you very much for your detailed and constructive comments. We’re glad that the clarification on the state-action dependence addressed your main concern, and we will revise the corresponding claim in the final version.
> >
> > We also appreciate your feedback regarding oracle assumptions and large sized RL. While a detailed treatment of these issues is beyond the current scope, we agree they are important considerations. We will briefly discuss them in the final version as limitations and interesting directions for future work.
> >
> > Thank you again for your helpful suggestions and for your positive assessment of our work.

---

### Official Review · Reviewer_6EBS · 2025-07-03

**Clarity:** 3
**Significance:** 3
**Originality:** 3
**Rating:** 5
**Confidence:** 3

**Summary:**

The paper studies the problem where the RL agent is provided with a K-length sequence of forecasts about the future evolution of the state (dynamics). To deal with the tractability of state augmentation, a Bayesian value function approach called BOLA is derived, which marginalizes out the forecast, avoiding the exponential blowup in compute. Analysis of the Bellman-Jensen gap provides an interpretable performance gap of BOLA in terms of horizon K, prediction error of the forecast, and partial action predictability, and sample complexity bounds are also derived. Analysis of the method on wind farm storage control show significant cost savings with longer forecasts, and some robustness to imprecise forecasting.

**Questions:**

1. Can other assumptions on forecasts, like parametric transition distributions, be accommodated within the current framework, perhaps as part of a broad spectrum of assumptions with parametric model on one extreme and full transition matrix forecast on the other?

2. Does the framework support continuous or large state spaces (perhaps connected to question 1 above)? What modifications to BOLA will be needed to avoid the blowup in compute with value iteration?

**Ethical Concerns:**

["NO or VERY MINOR ethics concerns only"]

**Final Justification:**

- Most of my concerns were centred on addressing the generality and necessary assumptions of the proposed framework. These issues have been clarified with examples in the rebuttal.
- Another aspect was clarification of the novelty of the theory in relation to prior work which was addressed well by the authors.
- The error regarding sample efficiency mentioned by another reviewer was also clarified.
- I have therefore decided to maintain my current score of Accept because the paper is technically strong and innovative.

**Limitations:**

I think more discussion should be provided about scalability and how to avoid the blowup in compute for larger state/action spaces. Additional experimentation would be quite useful to assess how well BOLA works across different types of problems.

**Quality:**

3

**Strengths And Weaknesses:**

Strengths:

1. The approach of embedding forecasts into the dynamic programming framework is quite elegant, and the Bellman equation shows how the forecast is utilized in downstream calculation.
2. The paper presents strong theoretical results in terms of performance gaps and sample efficiency.
3. The empirical evaluation is simple but demonstrates the use case and advantages quite clearly.

Weaknesses and suggestions:

1. In many practical settings, it may not be ideal to forecast each component of the transition matrix independently as described by sigma_k. Instead, many domain-specific models, such as the financial forecasting example mentioned in the preamble as well as many OR-specific problems like inventory control, queueing, etc. could often be parametric (i.e. Black-Scholes model). Other forecasts could perhaps be extrapolated from data using a neural network or other parametric model. It is not clear whether the current framework supports such use cases or how to adapt it.

2. The analysis on the Bellman-Jensen gap is quite similar to the work of Mercier et. al. [1] on anticipatory sampling, which has been widely known in stochastic planning. It might be good to cite this (and related) work, and ideally discuss the main limitations of their analysis on the so-called "anticipatory gap" and why it is difficult to apply directly towards forecasts.

3. (10)-(11) and value iteration might work well for small state/action/forecast spaces, but I don't think this will scale to problems of significant size. While I understand the core of the paper is theory-focused, an experiment on a larger version of the current problem with function approximation could significantly strengthen the paper's empirical evaluation. I understand that (6) might be applied within an existing model-based (deep) RL algorithm, but it might be good to discuss any practical limitations or necessary modifications to the current algorithm to scale up the current approach.

[1] Mercier, Luc, and Pascal Van Hentenryck. "Performance Analysis of Online Anticipatory Algorithms for Large Multistage Stochastic Integer Programs." IJCAI. 2007.

---

> ### Author Rebuttal · Authors · 2025-07-30
>
> We thank the reviewer for the detailed and thoughtful review. We're encouraged by your positive evaluation of our theoretical contributions and the clarity of our framework. We also appreciate your constructive suggestions. We address each of these points below.
>
>
> **Q1. Can other assumptions on forecasts, like parametric transition distributions, be accommodated within the current framework, perhaps as part of a broad spectrum of assumptions with parametric model on one extreme and full transition matrix forecast on the other?**
>
>
> **Response:**  Thank you for raising this insightful point. Our framework is flexible enough to accommodate a wide range of forecasting assumptions from full transition matrices to **parametric models**. As long as the forecast can be expressed in terms of (or mapped to) some transition probabilities, either explicitly or via a compact parameterization, it can be seamlessly integrated into our approach.
>
> * As the reviewer suggests, many real-world systems such as queueing and inventory control exhibit structured dynamics that can be characterized using only a few parameters. For example, in the classic discrete-time M/M/1 queue, the entire transition matrix can be fully determined by forecasting just two parameters: the arrival probability $\lambda$ and the service probability $\mu$. This allows us to compute all probabilities like $P(s_{t+1} = j+1 \mid s_t = j) = \lambda(1 - \mu)$ (i.e., an arrival occurs but no service), without needing to explicitly give the full matrix.
>
> * Moreover, the required forecast model can be incomplete. our framework also supports **partial forecasts** with predictable action set $\mathcal{A}^-$, where only a subset of transitions can be predicted.
>
> **Q2. Does the framework support continuous or large state spaces (perhaps connected to question 1 above)? What modifications to BOLA will be needed to avoid the blowup in compute with value iteration?**
>
>
> **Response:**   Yes, our framework can indeed be extended to large or even continuous state settings. The key is to learn the high-dimensional Bayesian value function $V^{{Bayes}}(s)$. To achieve this, we can use function approximation (e.g., neural networks) to learn $V^{Bayes}(s)$ via fitted value iteration. It is achievable because the Bayesian value function $V^{\text{Bayes}}(s)$ has the same dimensionality and quite similar definition (only different in prediction $\sigma$) of value function in standard MDPs. With the learned Bayesian value function, we can calculate the optimal policy follow a similar routine.
>
> At a high level, our method remains applicable in more general settings because it preserves similar fundamental notions of Bellman optimality condition and optimal policies shared with classical MDPs.
>
> **Q3. The analysis on the Bellman-Jensen gap seems similar to the work of Mercier et. al. [1] on anticipatory sampling, which has been widely known in stochastic planning. It might be good to cite this (and related) work, and ideally discuss the main limitations of their analysis on the so-called "anticipatory gap" and why it is difficult to apply directly towards forecasts.**
>
> **Response:** Thank you for the helpful suggestion. The anticipatory gap studied by Mercier et al. [1] is indeed closely related to our Bellman-Jensen gap when the prediction window is limited to $K=1$. However, our setting is significantly more general and challenging, in several key ways:
>
> * We allow arbitrary prediction windows $K \geq 1$;
> * We consider maximizing infinite-horizon cumulative reward objectives, not just terminal rewards;
> * We allow for incomplete predictions regarding subset $\mathcal{A}^-$ and prediction errors.
>
> Moreover, the analysis of the gap in [1] is primarily **case-based and lacks general theoretical characterization**. As acknowledged by the authors themselves, “We are currently applying this method on more complex problems but proofs quickly become very cumbersome. As an alternative, we discuss another, empirical way to argue that the GAG of a problem is small”, they resort to empirical justification rather than formal analysis.
>
>
> In contrast, our work provides a **general, explicit bound on the Bellman-Jensen gap for arbitrary MDPs and any prediction window $K$**. A key technical challenge lies in characterizing the recursive sum of coupled Jensen gaps across multi-step transitions. To address this, we introduce a novel dyadic horizon decomposition and a variance-based analysis (both in Appendix E) that enables us to decouple the multi-step dependencies in the Bellman operator.
>
> Analytically quantifying the value of prediction is a long-standing challenge, especially for unstructured problems. Most existing results rely on strong assumptions—e.g., linear dynamics as in LQR settings [2]. To our knowledge, our work provides the first general theoretical analysis of the value of prediction in MDPs with arbitrary forecast lengths and incomplete information.
>
> >[1] Mercier, Luc, and Pascal Van Hentenryck. "Performance Analysis of Online Anticipatory Algorithms for Large Multistage Stochastic Integer Programs." IJCAI. 2007.
> >[2] Yu, C., Shi, G., Chung, S.-J., Yue, Y., and Wierman, A. (2021). The power of predictions in online control. Advances in Neural Information Processing Systems, 33:1994–2004

---

> ### Comment · Reviewer_6EBS · 2025-08-05
> **Rebuttal**
>
> Dear authors,
>
> Thank you for clarifying the points in the rebuttal.
>
> In particular, I now understand the limitations of the Mercier et al., work and indeed I agree that the current framework and theoretical analysis applies in more general settings.
>
> I think the current paper is technically solid, and I have learned a lot from the manuscript and subsequent discussions. so I will maintain my current score.
>
> I think clarification of the most important points discussed here would significantly strengthen the overall write up, in addition to the asymptotic analysis clarification suggested by reviewer jBsU:
> 1. The learnability aspects and asymptotic bias arguments raised by reviewer DyeW and in the rebuttal are quite interesting. I wonder whether this is also related to the idea of an "uninformative action" in MDP parameter learning? Specifically, in your queueing example, it is impossible to acquire any new information about the arrival rate if no customers are admitted to the queue. Often dealing with such settings requires pretty strong assumptions on the structure of the MDP w.r.t which the current work is somewhat agnostic to. I think it would be interesting to discuss what the expected result of the algorithm and potential limitations would be.
> 2. The parameter learning setting as discussed still seems more practically useful, particularly in large state/action spaces. I think including this example up front, and more generally, common uses cases/problems the current framework was designed to handle, would strengthen the writeup.
>
> Once again, thank you for addressing the points in your rebuttal.

---

> > ### Author Response · Authors · 2025-08-06
> >
> > Thank you very much for your encouraging and thoughtful comments.
> >
> > We are glad that the clarification helped distinguish our contribution from Mercier et al., and we appreciate your recognition that the proposed framework applies more generally. We also appreciate your suggestions regarding learnability and asymptotic bias. As you pointed out, in some settings such as the queueing example, certain actions (e.g., never admitting customers) may prevent relevant environment parameters (e.g., arrival rates) from being revealed. While this issue primarily concerns the exploration policy used during model learning, our framework focuses on analyzing value-based planning given access to an oracle or a learned model. We will clarify this distinction in the final version, and note that characterizing learnability under limited exploration is an important direction for future work.
> >
> > We also agree with your suggestion to better highlight the motivating use cases. We will revise the introduction to bring the queueing example forward and more clearly identify the broader class of problems our framework is designed to address, such as parameterized MDPs with large or structured state/action spaces. Thank you for pointing out additional practical scenarios, which further illustrate the generality and practical relevance of our approach in real-world settings.
> >
> > Thank you again for your constructive and insightful feedback.

---

### Official Review · Reviewer_DyeW · 2025-07-07

**Clarity:** 3
**Significance:** 3
**Originality:** 3
**Rating:** 5
**Confidence:** 4

**Summary:**

The authors study the setting of RL in MDPs where additional (imperfect) predictions of multi-step predictions are given. They formalize a setting for MDPs with multi-step transition predictions. They define the *Bayesian value function* for prediction-aware decision-making, and work out the Bellman optimality equations for this value function. Towards analyzing the *value* of imperfect predictions, they give a bound on the difference between the Bayesian (imperfect) value function and the value function given perfect information. They lastly introduce a tabular model-based algorithm (BOLA) that estimates a model and incorporates predictions to compute the optimal Bayesian policy, with sample complexity guarantees. Experiments are run.

**Questions:**

See above.

**Ethical Concerns:**

["NO or VERY MINOR ethics concerns only"]

**Limitations:**

Yes.

**Quality:**

3

**Strengths And Weaknesses:**

This is an interesting paper on incorporating imperfect predictions into the RL problem. The formalism is clean, and the Bayesian value function approach feels like the correct solution concept. The theoretical results, while restricted to the tabular setting, are clean and neatly demonstrate the value of imperfect predictions. I am not aware of related work that overlaps too closely with this one. The writing is polished and brings out the key intuitions. I have skimmed the proofs and they seem correct. The experimental results on wind-farm storage are cool.

I have no major complaints, and just some minor points of confusion that I would appreciate some clarification on:
1. Equation 1 should be $P(s' \mid s,a,\sigma^\star_k)$?
2. What does it mean to condition on $\sigma$ when $\sigma = \sigma^\star + \varepsilon$ is imperfect (for example, in the expression of Equation 5.) Do we treat $\sigma$ here as $\sigma^\star$, ignoring the imperfection? In general it's not crystal clear what it means to condition on $\sigma$  (e.g. even in the value function definition of Equation 3). Can the authors be more formal about the probability kernels involved?
3. The current algorithm 1 samples transitions and predictions and does a plug-in estimator to learn \hat{V}^{Bayes}_{\varepsilon} that approximates V^{Bayes}_{\varpesilon}. Intuition suggests that sampling more $\sigma$ variables could eventually give us a good approximation to $\sigma^\star$ (by concentration and since $\sigma = \sigma^\star + \varepsilon$) and we should thus be learning V^{Bayes}_0. What is happening here?

Two additional questions:
1. It feels like the BOLA algorithm can be extended to settings beyond the tabular one. At a high level the two components involve: estimating $P(\sigma)$, and estimating the rest of the model ($r$ and $P(s' | s,a)$). Can we indeed say that this methodology works in any setting where the above quantities are learnable?
2. There is a trade-off here between the statistical and computational costs of learning transitions over $K$ steps (which gets more expensive due to an exponential scaling in $K$) and the quality of the learned policy (since more steps gives a better policy). Is there any interesting way to adaptively choose $K$ to balance these considerations?

---

> ### Author Rebuttal · Authors · 2025-07-30
>
> Thank you very much for your positive and thoughtful review of our paper. We greatly appreciate your recognition regarding our novelty, theoretical results, experiments, and writing. Below, we address your minor points of confusion:
>
> **Q1. Equation 1 should be $P(s'|s,a,\sigma_k^*)$?**
>
> **Response:** Yes, that is correct. Thank you for catching this typo!
>
> **Q2. What does it mean to condition on $\sigma$ when $\sigma = {\sigma}^* +\epsilon$ is imperfect (for example, in the expression of Equation 5.) Do we treat $\sigma$ here as $\sigma^*$, ignoring the imperfection? In general it's not crystal clear what it means to condition on $\sigma$ (e.g. even in the value function definition of Equation 3). Can the authors be more formal about the probability kernels involved?**
>
> **Response:** Thank you for raising this point. In our framework, we condition on the imperfect transition prediction $\sigma$, not the perfect one ${\sigma}^* $. Specifically, throughout the paper (including in Equations 3 and 5), we use the imperfect transition probabilities defined as $P(s'|s,a,\sigma_k) = \sigma_k((s,a),s')$. This ensures that the agent makes decisions based solely on the imperfect predictions available to it (the agent never observes the perfect transitions $\sigma^*$). We agree that the confusion may have arisen due to the typo in Equation 1.
>
> **Q3. The current algorithm 1 samples transitions and predictions and does a plug-in estimator to learn hat{V}^{Bayes}_epsilon that approximates V^{Bayes}_epsilon. Intuition suggests that sampling more variables could eventually give us a good approximation to accurate transition prediction, because by concentration, $\sigma = \sigma^{*} + \epsilon$, and we should thus be learning unbiased value function. What is happening here?**
>
> **Response:**  Thank you for the question. In our approach, we sample imperfect transition predictions $\sigma$ to approximate their distribution $f_{\sigma}$. This distribution reflects forecast errors of size $\epsilon$, and is different from the true distribution $f_*$ over accurate transitions $\sigma^*$.
>
> Importantly, any accurate prediction $\sigma^*$ is a deterministic (binary) transition matrix, where each row is a one-hot vector. In contrast, the noisy forecasts $\sigma$ may contain non-binary probabilities (like 0.1, 0.9) due to errors. Thus, even with many samples from $f_{\sigma}$, we do not recover the distribution of accurate predictions. Hence, the learned Bayesian value function will converge, but to a biased solution. This is different from using transition samples to recover the accurate transition kernel using empirical average.
>
> **Two additional questions:**
>
>
> **Q4. It feels like the BOLA algorithm can be extended to settings beyond the tabular one. At a high level the two components involve: estimating $P(\sigma)$, and estimating the rest of the model (r and $P(\sigma)$). Can we indeed say that this methodology works in any setting where the above quantities are learnable?**
>
> **Response:**  Yes, the two-stage BOLA algorithm can indeed be extended beyond the tabular setting, as long as $P(\sigma)$ and other relevant system models (such as $r$ and $P(\sigma)$) are learnable. To extend to continuous state settings, we can use function approximation (e.g., neural networks) to learn the continuous Bayesian value function $V^{\text{Bayes}}(s)$. This can be done via fitted value iteration, leveraging the structural similarity between our Bayesian Bellman equation and that of standard MDPs.
>
> At a high level, BOLA remains applicable in more general settings because it preserves the fundamental notions of value functions and optimal policies shared with classical MDPs.
>
> **Q5. There is a trade-off here between the statistical and computational costs of learning transitions over K steps (which gets more expensive due to an exponential scaling in K) and the quality of the learned policy (since more steps gives a better policy). Is there any interesting way to adaptively choose K to balance these considerations?**
>
> **Response:**  This is an excellent question. As shown by both Theorem 4.1 and our experiments, the marginal benefit of extending the prediction horizon $K$ decreases exponentially, while the computational cost increases exponentially. To handle this trade-off, a simple yet effective strategy is to incrementally increase $K$ (e.g., starting from $K=1$) until the computational budget is met. Since the total trial cost $\sum_{k=1}^{K} D^k \approx D^K$ (where $D^k$ is the computational cost with prediction window $k$), this approach naturally identifies the optimal $K$ that balances reward performance and computational cost, and is nearly optimal in terms of total trial cost. In practice, this greedy search often achieves a good performance.

---

> > ### Comment · Reviewer_DyeW · 2025-08-07
> > **Thanks for your reply**
> >
> > Dear authors,
> >
> > Thank you for your reply. I have no further comments at this time and maintain my positive score.

---

> > > ### Author Response · Authors · 2025-08-07
> > > **Thanks for your comments**
> > >
> > > Thank you so much for your constructive suggestions and for maintaining your positive evaluation. We greatly appreciate your thoughtful feedback and are glad that no further justifications are needed at this stage.

---

### Author Response · Authors · 2025-08-07
**General response to all reviewers**

We thank all the reviewers for their careful reading of our paper and their thoughtful, constructive feedback. We are encouraged by the positive assessments and the engaging discussions. Below we provide a general response to highlight the key insights and contributions of our work.

**Key Contributions**:
This work is motivated by a fundamental question in reinforcement learning: how can one leverage predictions of future transitions to improve decision-making? Actually, this is a long-standing problem for online decision-making, especially for problems without special structures like MDPs (As discussed with Reviewer 6EBS in Q3). While the use of predictive models is common in practice, a precise understanding of their algorithmic value has remained unclear.

**Surprisingly, we find that incorporating predictions into reinforcement learning reveals a recursive structure in the value function. This structure gives rise to an infinite sequence of nested local Jensen gaps, which emerge from the reordering of the $\max$ and $\mathbb{E}$ operators.** This insight enables a dyadic decomposition of the planning horizon, where we construct an infinite sequence of exponentially increasing prediction windows to characterize the cumulative value improvement. This supports a variance-based analysis showing that each doubling of the window yields an exponentially smaller gain in value. **Notably, such analysis is technically challenging and addresses key limitations of prior literature, as discussed with Reviewer 6EBS.**


We develop practical algorithms based on these ideas and demonstrate that the framework remains effective even with imperfect or incomplete predictions. Notably, due to the high dimensionality of transition predictions, directly incorporating them into the classical RL paradigm exponentially increases the effective state-action space (more details are discussed in Section 3 of the manuscript, line 118-124). This typically makes the learning problem more challenging and could lead to worse sample efficiency. **Counter-intuitively, as discussed with Reviewer jBsU, our framework achieves better sample efficiency by leveraging the structure introduced by predictive information**.



**To our knowledge, this is among the first attempts to systematically characterize the impact of general transition predictions on reinforcement learning. We are excited by the potential it offers for developing principled approaches to learning and decision-making under predictive uncertainty.**

---

### Decision · Program_Chairs · 2025-09-17

**Decision:**

Accept (spotlight)

**Comment:**

The paper studies whether (and how) future-state multistep predictions can be integrated into an RL process and how this impacts the overall performance.

The question studied in the paper is fundamental in nature and particularly significant in domains where (multi-step) world models are available. The contribution of the paper is shading light on this question is extensive: from the formalization of the question, to development of an algorithm and its theoretical properties, all the way to empirical validation in a small-scale but practically relevant problem (wind-farm storage control task). While the reviewers pointed out some weaknesses (e.g., validity of assumptions, limitation to tabular MDPs for the theory, comparison with existing literature, and small-scale experiments), these were clearly out-weighted by the significance of the topic and the novelty of the results, which position the paper to have the potential to inspire further developments in this direction. Based on this, there was a clear consensus towards acceptance.

I would still encourage the authors to properly integrate the rebuttal (e.g., comparison to Mercier et. al., clarification of the theoretical results) and expand as much as possible on the path towards practical algorithms based on BOLA.